# TDP-43 interacts with amyloid-β, inhibits fibrillization, and worsens pathology in a model of Alzheimer's disease

Yao-Hsiang Shih [1,2,3,8], Ling-Hsien Tu [1,4,8], Ting-Yu Chang[1,5], Kiruthika Ganesan[1], Wei-Wei Chang[1], Pao-Sheng Chang [1], Yu-Sheng Fang[1,6], Yeh-Tung Lin[1,5], Lee-Way Jin[7] & Yun-Ru Chen [1,5,6✉]

TDP-43 inclusions are found in many Alzheimer's disease (AD) patients presenting faster disease progression and greater brain atrophy. Previously, we showed full-length TDP-43 forms spherical oligomers and perturbs amyloid-β (Aβ) fibrillization. To elucidate the role of TDP-43 in AD, here, we examined the effect of TDP-43 in Aβ aggregation and the attributed toxicity in mouse models. We found TDP-43 inhibited Aβ fibrillization at initial and oligomeric stages. Aβ fibrillization was delayed specifically in the presence of N-terminal domain containing TDP-43 variants, while C-terminal TDP-43 was not essential for Aβ interaction. TDP-43 significantly enhanced Aβ's ability to impair long-term potentiation and, upon intra-hippocampal injection, caused spatial memory deficit. Following injection to AD transgenic mice, TDP-43 induced inflammation, interacted with Aβ, and exacerbated AD-like pathology. TDP-43 oligomers mostly colocalized with intracellular Aβ in the brain of AD patients. We conclude that TDP-43 inhibits Aβ fibrillization through its interaction with Aβ and exacerbates AD pathology.

[1] Genomics Research Center, Academia Sinica, 128, Academia Road, Section 2, Nankang District, Taipei 115, Taiwan. [2] Department of Anatomy, School of Medicine, Kaohsiung Medical University, 100, Shih-Chuan 1st Road, Sanmin District, Kaohsiung 80708, Taiwan. [3] Department of Medical Research, Kaohsiung Medical University Hospital, 100, Tzyou 1st Road, Sanmin District, Kaohsiung 80756, Taiwan. [4] Department of Chemistry, National Taiwan Normal University, No. 88, Section 4, Ting-Chow Road, Taipei 11677, Taiwan. [5] Department of Biochemical Science and Technology, National Taiwan University, No. 1, Section 4, Roosevelt Road, Taipei 10617, Taiwan. [6] Institute of Bioinformatics and Structural Biology, National Tsing Hua University, 101, Kuang fu Road, Section 2, Hsinchu 30013, Taiwan. [7] Department of Pathology and Laboratory Medicine, Alzheimer's Disease Center, 2805 50th Street, University of California Davis Medical Center, Sacramento, CA 95817, USA. [8] These authors contributed equally: Yao-Hsiang Shih, Ling-Hsien Tu. ✉email: yrchen@gate.sinica.edu.tw

Alzheimer's disease (AD) is the most common type of dementia in which the two pathological hallmarks are senile plaques comprising amyloid-β (Aβ) and neurofibrillary tangles composed of hyperphosphorylated tau[1,2]. Aβ, mainly consists of 40 or 42 amino acids, is a proteolytic cleavage product of amyloid precursor protein by sequential cleavages of β- and γ-secretases. Aβ42 is considered more toxic as evidenced from both in vitro[3] and in vivo studies[4], where Aβ42 to Aβ40 ratio implicated in some familial AD cases is also important[5,6]. Aβ spontaneously aggregates into amyloid fibrils with a cross β-sheet structure, where prefibrillar Aβ oligomers are considered the most toxic species and attributed to neuronal dysfunction and cognitive impairment[7]. Amyloid fibrillization adopts a nucleation-dependent polymerization mechanism in which amyloid fibrils serve as a nucleus to seed soluble species and skip the nucleation[8]. Aβ deposits extracellularly to form amyloid plaques, but also accumulate intracellularly as shown in human patients and transgenic mice[9].

TDP-43 pathology was first found in cytosolic inclusions in patients with amyotrophic lateral sclerosis (ALS) and frontotemporal lobar degeneration (FTLD-TDP)[10]. TDP-43 inclusions were also reported in the brain of AD patients with a prevalence of ~30 to ~57%[11–13]. The AD patients with TDP-43 inclusions have more severe memory loss and hippocampal atrophy[13,14]. The highly significant pathological role of TDP-43 is further suggested by a recently proposed diagnostic entity called limbic-predominant age-related TDP-43 encephalopathy (LATE), which predominantly occurs in cognitively impaired oldest-old individuals with co-existing medial temporal amyloid plaques and tauopathy[15]. TDP-43 is a highly conserved RNA/DNA-binding nuclear protein consisting of 414 amino acids with a molecular mass of 44,740 Da[16]. Full-length TDP-43 is constitutively expressed in many tissues, including brain, heart, lung, kidney, and muscle[17,18]. TDP-43 contains two RNA binding domains (RRM1 and RRM2), which are flanked by an N-terminal domain and a C-terminal glycine-rich region. The N-terminal region was shown to dimerize that facilitates nucleotide binding and splicing[19,20], whereas, C-terminal regions are highly prone to aggregation[21]. Recombinant full-length TDP-43 forms stable spherical oligomers, which are toxic to neurons in primary culture and in mouse brain following intrahippocampal injection[22]. In our previous study, we generated a conformational-specific polyclonal antibody namely poly TDP-O that specifically recognizes TDP-43 oligomers and used it to confirm the existence of TDP-43 oligomers in the brain of patients with FTLD-TDP[22].

Despite the common occurrence of TDP-43 proteinopathy in AD, the pathological role of TDP-43 in AD is still largely unknown. A study has shown that in a transgenic AD mouse model, 3xTg, which develops both amyloid plaques and abnormal tau, the levels of mouse TDP-43 and its C-terminal fragment were significantly increased and positively correlated with the accumulation of Aβ oligomers[23]. In lentiviral TDP-43-injected rat, TDP-43 expression induced neuroinflammation and increased β-secretase activity, accelerating the production of Aβ and APP-C-terminal fragments[24]. Previously, we have shown that full-length TDP-43 forms toxic spherical oligomers readily (Supplementary Fig. 1a) and can transform Aβ to amyloid oligomers[22]. However, amyloid plaques and TDP-43 inclusions are not colocalized in the brain of AD patients[25].

To elucidate the role of TDP-43 in AD, here we aimed to characterize the interaction between TDP-43 and Aβ and investigated the effect of TDP-43 in AD pathology. We examined the effect of TDP-43 during Aβ aggregation and characterized the interaction by different TDP-43 variants. The toxicity of TDP-43-induced Aβ species was examined by electrophysiology and intrahippocampal injection to WT mice for memory behavior.

We further intrahippocampally injected TDP-43 in transgenic AD mice and showed that TDP-43 worsened the spatial memory and AD pathology of mice. Finally, we showed that TDP-43 oligomers mostly colocalized with intracellular Aβ in AD brain tissues.

## Results

**TDP-43 inhibits Aβ fibrillization at the initial and oligomeric stages of Aβ, does not reverse the Aβ fibril state, and has no impact on Aβ seeding reaction.** To understand the effect of TDP-43 on Aβ aggregation, we first characterized Aβ aggregation with and without TDP-43 by multiple biochemical techniques. Aβ40 was used for all biochemical assays except for additional indication due to the relatively easy handling. First, we conducted thioflavin-T (ThT) assay to monitor the fibrillization kinetics of Aβ (Fig. 1a). ThT can specifically bind to cross-β-sheet structures in amyloid fibrils and emit fluorescence[26]. TDP-43 stock or buffer control was mixed with Aβ solution, and the final Aβ and TDP-43 concentrations were 25 and 0.25 μM, respectively. The ThT results showed that Aβ alone adopted a classic amyloid fibrillization pathway with a nucleation phase (lag time), an elongation phase, and a plateau phase. In the absence of TDP-43, Aβ possessed a lag time of ~25 h, whereas the lag time was significantly prolonged to ~75 h in the presence of TDP-43. The final ThT signal of Aβ with TDP-43 was reduced to ~50% compared with the signal of Aβ alone. No ThT signal was found in TDP-43 alone. Next, we employed two conformation-specific antibodies, namely A11 and OC, to facilitate the understanding of the aggregation process (Fig. 1b). The time-course samples of Aβ with and without TDP-43 were subjected to dot blot by using A11 and OC antibodies. A11 recognizes prefibrillar oligomers[27], and OC specifically recognizes fibrillary oligomers and amyloid fibrils[28,29]. At the experimental concentration, TDP-43 did not show A11 and OC signals (Supplementary Fig. 1b). During the incubation, Aβ alone aggregated into oligomers and gradually formed fibrils. Both A11 and OC signals were significantly enhanced at 22 h and saturated after 46 h. In the presence of TDP-43, the A11 signal did not change; however, the OC signal showed delayed saturation at 70 h rather than 46 h, indicating that TDP-43 delays Aβ fibril formation. The conformational changes in Aβ were monitored at different time points through far-UV circular dichroism (CD; Fig. 1c). Aβ in the absence or presence of TDP-43 was observed, and the spectra were subtracted with buffer or TDP-43 background. The CD result showed that Aβ initially appeared as a random coil with a minimal signal at 198 nm, gradually formed a β-sheet at ~218 nm from 22 h to 96 h, and saturated at 118 h. In the presence of TDP-43, Aβ retained its random coil state up to ~46 h and changed to a β-sheet structure from 70 h to 118 h. This result is consistent with our observation in the ThT and dot blot assays. We observed the morphology of the aggregates through transmission electron microscopy (TEM) (Fig. 1d). At the end of aggregation (>160 h), Aβ alone formed long, and extensive amyloid fibrils, whereas, Aβ in the presence of TDP-43 formed less dense and shorter filaments.

To understand how TDP-43 hinders Aβ aggregation, we added TDP-43 to different Aβ aggregation states and monitored the ThT signals. Aβ monomers, oligomers, and preformed fibrils were prepared and validated via TEM (Supplementary Fig. 2a–c). In this set of experiments, the concentrations of Aβ and TDP-43 were 25 and 0.25 μM, respectively. Consistent with previous results, TDP-43 added to Aβ monomer effectively blocked Aβ fibrillization (Fig. 1e). When TDP-43 was added to the Aβ oligomer solution (Fig. 1f), it is still able to potently inhibit Aβ fibrillization even Aβ already formed oligomeric species. When

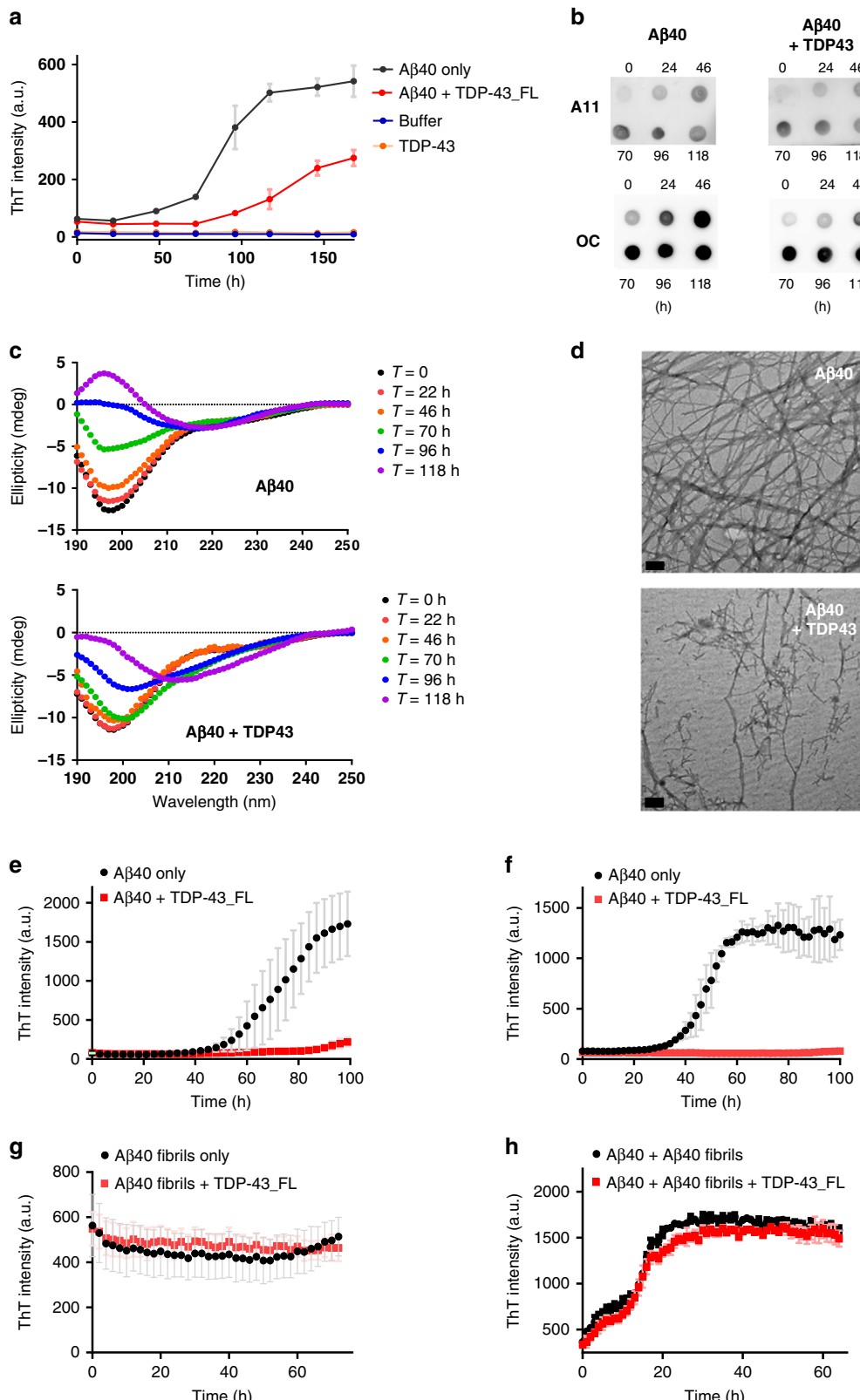

TDP-43 was introduced to the Aβ fibril solution (Fig. 1g), TDP-43 could not alter the status of Aβ fibrils. The ThT signal of Aβ fibrils remained the same during the observation. We also used immunogold TEM to observe the possible location of TDP-43 in Aβ fibrils by immunogold labeling of TDP-43. We found that some immunogold labeled TDP-43 attached to Aβ fibrils especially at the clusters of fibrils (Supplementary Fig. 2d), but no apparent morphological changes of Aβ fibrils were observed. Also, we wonder if TDP-43 could block the seeding event of Aβ fibrillization. Aβ monomer was added with 10% preformed Aβ fibril seeds simultaneously with and without TDP-43. The seeds were prepared from preformed 25 μM Aβ fibrils after sonication. The results showed that Aβ fibrillization kinetics with the seeding remained the same with or without TDP-43 (Fig. 1h), indicating

**Fig. 1 TDP-43 inhibits Aβ fibrillization at initial and oligomer stages of Aβ. a** Fibrillization kinetics of Aβ40 (black) and Aβ40 with TDP-43 (red) monitored by ThT assay in 10 mM Tris buffer, pH 8.0. TDP-43 alone (orange) and buffer alone (blue) are also shown. The averaged data from three replicates and standard deviations are plotted. Aβ40 concentration was 25 μM and TDP-43 concentration was 0.25 μM. **b** Dot blotting of the time point samples by anti-amyloid oligomers A11 antibody and anti-amyloid fibrils OC antibody. **c** Far-UV CD spectra of Aβ40 in the absence and presence of TDP-43_FL after 0 (black), 22 (red), 46 (orange), 70 (green), 96 (blue), and 118 (purple) h incubation. The buffer control and TDP-43 background were subtracted. **d** TEM images of the end-point product of Aβ40 alone in ThT assay, and the end-point product of Aβ40 with TDP-43. The scale bars are 100 nm. Two times of independent experiments were performed. **e–h** Fibrillization kinetics of Aβ40 from different stages with and without full-length TDP-43. Aβ40 monomer (**e**), Aβ40 oligomers (**f**), Aβ40 fibrils (**g**), and Aβ40 monomer in the presence of 10% fibril seeds (**h**) were incubated with (red) and without full-length TDP-43 (black) and monitored by ThT assay in 10 mM Tris buffer, pH 8.0. Aβ40 concentration was 25 μM, and TDP-43 concentration was 0.25 μM. Aβ fibril seeds were prepared from sonicated Aβ40 fibrils at 25 μM. The averaged data from three replicates and standard deviations are plotted. Source data are provided as a Source Data file.

that TDP-43 could not prevent Aβ fibril growth by seeding. Therefore, our result showed that TDP-43 could only inhibit Aβ fibrillization at the initial and oligomeric states, but not affect Aβ conformation at the fibril state and in the seeding reaction.

**N-terminal containing TDP-43 variants inhibit Aβ fibrillization.** TDP-43 contains four structural regions, which are an N-terminal domain, two RNA recognition motifs (RRM1 and RRM2), and a C-terminal glycine-rich region (Fig. 2a). TDP-43 may function as a homodimer via the interaction of N-terminal regions[20,30], but the misfolded structure of TDP-43 has not been clearly elucidated. In our previous study, full-length TDP-43 readily formed oligomers that were eluted in the void volume in size exclusion chromatography (SEC)[22]. Here, we aimed to clarify the prerequisite structural regions of TDP-43 for inhibiting Aβ fibrillization. We constructed and purified different structural regions of TDP-43, including TDP-43 with a C-terminal region truncated (named TDP-43_265, residue 1–265), an N-terminal region (named TDP-43_N-term, residue 1–100), and the RNA recognition region (named TDP-43_RRM1 + 2, residue 101-265). The calculated molecular weights of each variant were 32.5 kDa for TDP-43_265, 12.5 kDa for TDP-43_N-term, and 20.4 kDa for TDP-43_RRM1 + 2. The assembly of these truncated proteins was first characterized in SEC (Supplementary Fig. 3a). According to the molecular weight standards, TDP-43_265 and TDP-43_N-term formed homodimers with estimated molecular masses of 53 and 23 kDa, but TDP-43_RRM1 + 2 remained monomeric with an estimated molecular mass of 21 kDa. No aggregates in these TDP-43 variants were observed in SEC. We subjected the variants to dot blot and found that only full-length TDP-43 was immunopositive and that TDP-43_265 was very weakly immunopositive when probed by our polyclonal TDP-43 oligomer-specific antibody TDP-O[22] (Supplementary Fig. 3b).

Next, we tested the potency of TDP-43 variants in inhibiting Aβ fibrillization (Fig. 2b). As expected, full-length TDP-43 inhibited the formation of Aβ fibrils by prolonging the lag time and lowering the ThT fluorescence to 60% at the final time point. Surprisingly, TDP-43_265 and TDP-43_N-term exhibited strong inhibition on Aβ fibrillization. TDP-43_N-term prolonged the lag time of Aβ from 50 h to ~90 h and reduced ThT intensity to 75% compared with that of the Aβ only control, while TDP-43_265 prolonged the lag time from 50 h to 125 h, and reduced the ThT signal to ~9%. By contrast, TDP-43_RRM1 + 2 did not have much inhibitory effect on Aβ fibrillization, with the lag time of Aβ slightly prolonged from 50 h to 55 h, and the final ThT intensity slightly affected (decreased to ~90%). We employed TEM to observe the morphology of Aβ species at 100 and 150 h incubation (Fig. 2c). In the absence of TDP-43 variants, Aβ formed typical amyloid fibrils through time as expected. However, Aβ samples with full-length TDP-43, TDP-43_265, or TDP-43_N-term contained much less and shorter fibrils than Aβ

alone, whereas, Aβ with TDP-43_RRM1 + 2 still contained a significant number of fibrils.

The end-products of aggregation study were further examined by western blot (Fig. 2d, e). The samples collected after the end-point (150 h) of the ThT assay were stored at −20 °C and later subjected to sodium dodecyl sulfate polyacrylamide gel electrophoresis (SDS-PAGE). The samples were loaded onto Tris-Tricine SDS–PAGE and transferred onto a polyvinylidene fluoride (PVDF) membrane probed by the anti-Aβ antibody 6E10. We found that Aβ alone possessed a large amount of aggregates stocked on top of the gel (lane 5, Fig. 2d) so as Aβ with TDP-43 RRM1 + 2 (lane 4). Aβ incubated with full-length TDP-43 had few Aβ aggregates on top of the gel (lane 1). Aβ with TDP-43_265 and TDP-43_N-term (lane 2 and 3) had reduced aggregates compared with Aβ only. Considering that the aggregates stocked on top of the gel were not quantitative, we examined the lower portion of the SDA-PAGE. After enhancing the contrast of the low-molecular-weight species (Fig. 2e), we found that Aβ in the presence of full-length TDP-43 significantly enhanced Aβ monomer, dimer, and trimer (lane 1). Aβ in the presence of TDP-43_265 and TDP-43_N-term (lane 2 and 3) also showed a significant enhancement of monomer and dimer compared with Aβ alone (lane 5) and Aβ with TDP43_RRM1 + 2 (lane 4). Overall, the results demonstrated that full-length, residues 1–265, and N-terminal TDP-43 could inhibit Aβ fibrillization, leading to fibril reduction and low-molecular-weight species accumulation. We also tested the inhibitory effect of TDP-43 variants on Aβ42 fibrillization by ThT assay. The ThT result showed that full-length TDP-43 still significantly inhibited Aβ42 fibrillization by retarding the lag time from 20 h to 40 h (Supplementary Fig. 4). However, the other TDP-43 variants did not significantly affect Aβ42 fibrillization.

**TDP-43 variants interact with Aβ via different structural domains.** Considering that TDP-43 variants inhibited Aβ aggregation to different degrees, we examined the molecular interaction between TDP-43 variants and Aβ. We first employed enzyme-linked immunosorbent assay (ELISA) to detect the binding of Aβ to TDP-43 proteins (Fig. 3a). In this experiment, TDP-43 variants at 1 μM, 200 pmole, were immobilized onto the ELISA plate. N-terminal biotinylated Aβ40 was added at various concentrations ranging from 0 to 38 μM and subjected to strep-tavidin- horseradish peroxidase (HRP) detection. The ELISA results showed that the binding of Aβ was the strongest in the full-length TDP-43, followed by TDP-43_265 and TDP-43_RRM1 + 2. The weakest binding was found in the N-terminal TDP-43.

We further performed biolayer interferometry analysis to examine the on and off kinetic binding rates of TDP-43 variants (Fig. 3b–d). The biotinylated Aβ40 was immobilized onto streptavidin-coated sensors and then subjected to different concentrations of TDP-43 variants to detect the interaction

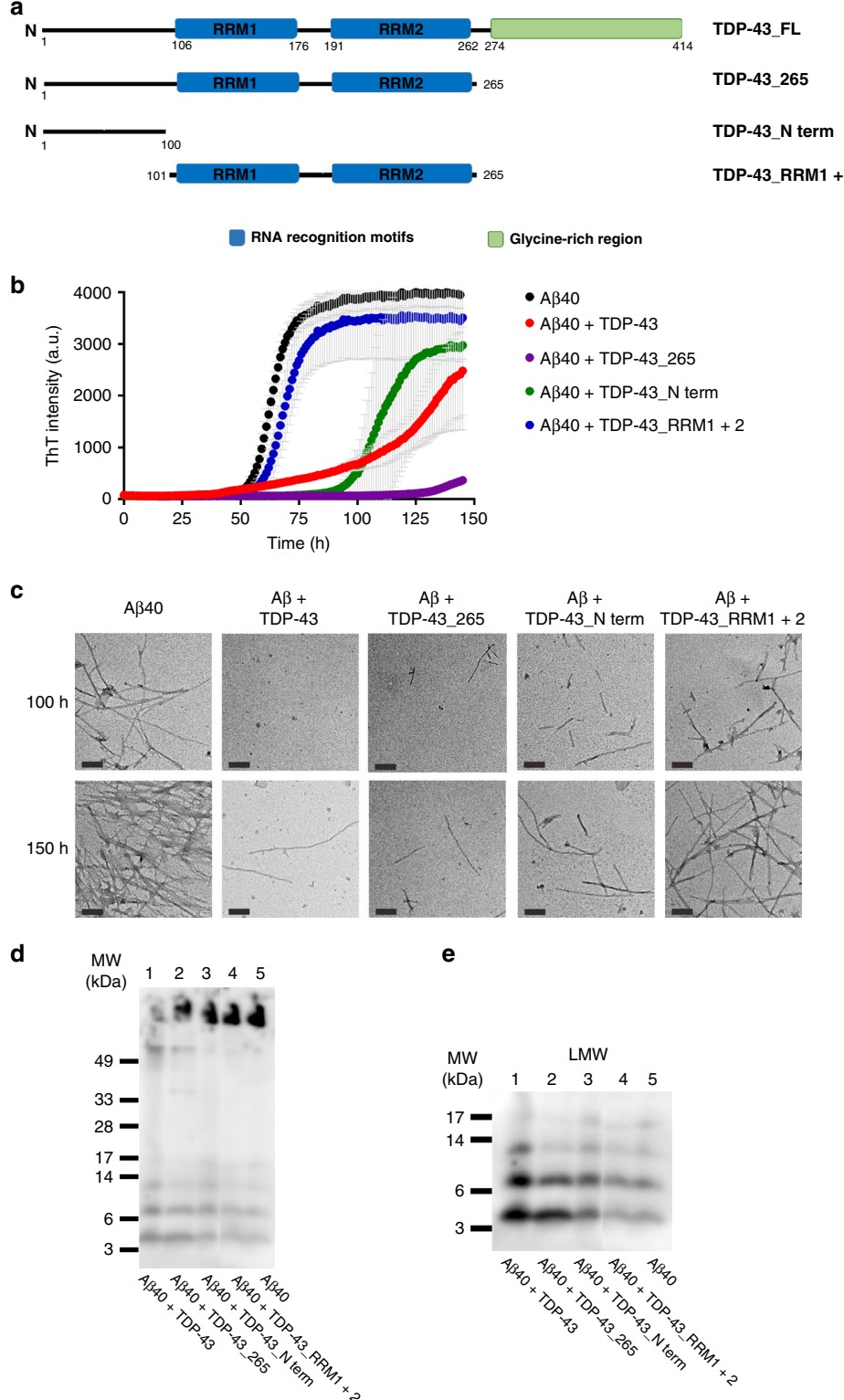

**Fig. 2 Truncated TDP-43 variants inhibit Aβ fibrillization. a** Illustration of the structural motifs of TDP-43. Full-length TDP-43 contains an N-terminal domain, two RNA recognition motifs (RRM1 and RRM2), and a glycine-rich region in its C-terminus. The constructs used in the study are shown. They are full-length TDP-43, TDP-43 aa 1–265 (TDP-43_265), TDP-43 aa 101–265 (TDP-43_RRM1 + 2), and TDP-43 aa 1–100 (TDP-43_N-term). **b** ThT assays of Aβ fibrillization (black) and with TDP-43 variants (TDP-43_FL, red; TDP-43_265, purple; TDP-43_N-term, green; TDP-43_RRM1 + 2, blue) in 10 mM Tris buffer, pH 8.0. Aβ40 concentration was 25 μM, and TDP-43 concentration was 0.25 μM. The averaged data from three replicates and standard deviations are plotted. **c** TEM images of Aβ species with and without TDP-43 proteins. Two incubation time points, 100 h and 150 h were chosen for examination. The samples were loaded on SDS-PAGE **d** and then probed by anti-Aβ antibody, 6E10 to show the relative amounts of Aβ40 amyloid fibrils. Two times of independent experiments were performed. **e** The enlarged area of Fig. 2d shows a clear band distribution of low-molecular-weight Aβ species. Source data are provided as a Source Data file.

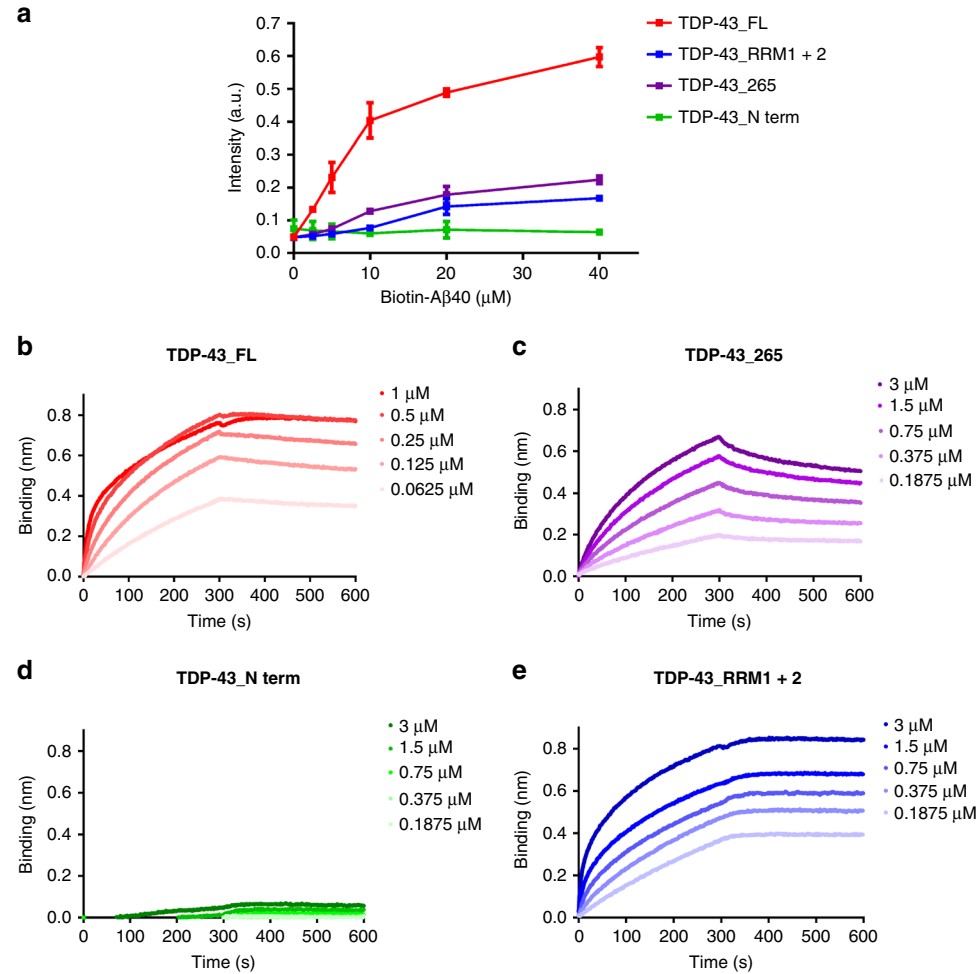

**Fig. 3 Interaction of TDP-43 variants and Aβ. a** Interaction of Aβ with TDP-43 variants was determined by ELISA. TDP-43 proteins were coated and detected by different concentrations of biotinylated Aβ. The averaged data from three replicates and standard deviations are plotted (TDP-43_FL, red; TDP-43_265, purple; TDP-43_N-term, green; TDP-43_RRM1 + 2, blue). **b–e** Binding response between Aβ40 and TDP-43 variants in biolayer interferometry analysis. TDP-43 at different concentrations (μM) were prepared and subjected to biolayer interferometry analysis by using Octet RED96 System (Pall ForeBio). The real-time binding response (nm) was measured in seconds. The loading amounts of Aβ40 are the same for all the experiments. Source data are provided as a Source Data file.

**Table 1 Affinity of TDP-43 variants and Aβ interaction. The rate constants $k_{on}$, $k_{off}$, and $K_D$ defined as $k_{off}/k_{on}$ were determined by Octet RED96 system to probe the interaction between TDP-43 and Aβ40.**

|  | $k_{on}$ (1/Ms) | $k_{off}$ (1/s) | $K_D$ (M) |
|---|---|---|---|
| TDP-43_FL | $3.4 \times 10^4$ | $<1.0 \times 10^{-7}$ | $<1.0 \times 10^{-12}$ |
| TDP-43_265 | $5.7 \times 10^3$ | $5.6 \times 10^{-4}$ | $9.9 \times 10^{-8}$ |
| TDP-43_N-term | N.D. | N.D. | N.D. |
| TDP-43_RRM1 + 2 | $9.0 \times 10^3$ | $<1.0 \times 10^{-7}$ | $<1.0 \times 10^{-12}$ |

N.D.: unable to be determined due to absence of significant binding signal.

between Aβ and TDP-43. Owing to the strong nonspecific binding of TDP-43_N-term to streptavidin-coated sensors, we added 0.005% Tween-20 in all experiments to reduce nonspecific binding of TDP-43. The sensograms were plotted, and $k_{on}$, $k_{off}$, and $K_D$ were calculated (Table 1). The parameters were successfully obtained for all TDP-43 variants except TDP-43 N-term. The $k_{on}$ rates for all other variants are around $10^3$–$10^4$ M/s. Both full-length TDP-43 and TDP-43_RRM1 + 2 have $k_{off}$ rate $<10^{-7}$/s indicating tight binding without easy dissociation. The

small $k_{off}$ rate resulted in a small dissociation constant for both TDP-43_FL and TDP-43_RRM1 + 2 ($K_D < 10^{-12}$ M). For TDP-43_265, the dissociation constant is $9.87 \times 10^{-8}$ M.

**TDP-43-induced Aβ species impair long-term-potentiation (LTP) in the hippocampus and the related spatial memory in vivo.** LTP is an important cellular mechanism related to learning and memory function[31]. To investigate whether TDP-43-induced changes in Aβ40 or Aβ42 aggregation impaired neuronal function, we measured Schaffer collateral-CA1 LTP in mouse brain slice (Fig. 4) after treatment of the aggregation samples as described in Fig. 1a. The brain slices were incubated with TDP-43-induced Aβ species, Aβ alone, TDP-43 alone, and buffer control for 30 min, and the LTP responses were measured after a theta burst stimulation. The Aβ40 experiments (Fig. 4a) showed that TDP-43-induced Aβ40 species significantly suppressed hippocampal LTP compared with that of the buffer control (repeated two-way ANOVA, treatment factor: $F = 28.21$, df 1/5, $p = 0.003$). However, no significant difference in the fEPSP slope was induced by the buffer control, Aβ40 alone (1 μM), and TDP-43 alone (0.01 μM) (repeated two-way ANOVA, treatment factor: $F = 1.21$, df 2/8, $p = 0.348$). In Aβ42 experiments, we prepared Aβ42 following previous literature[32] and incubated with

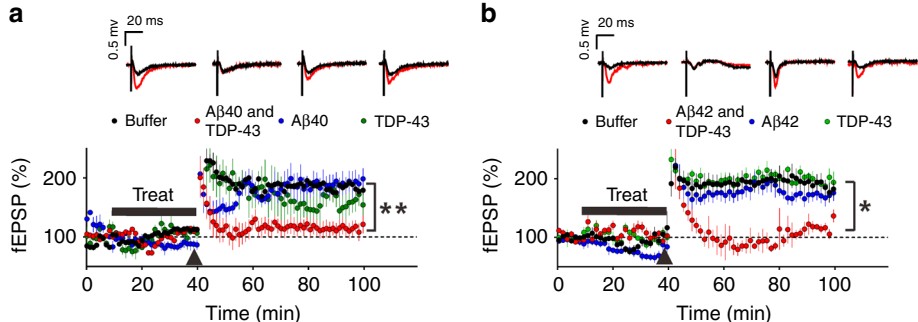

**Fig. 4 TDP-43-induced Aβ species impairs the hippocampus LTP.** Field EPSPs (fEPSPs) were measured from a Schaffer collateral fiber on a hippocampal slice from C57BL/6 mice. After the recording was stabilized for 10 min, the slices were treated with buffer, Aβ with TDP-43, Aβ alone, or TDP-43 alone for 30 min. After the treatment, the hippocampal slices were subjected to a theta burst stimulation (black arrowhead) to induce LTP. Scale bar, 0.5 mv, 20 ms. The averaged data and s.e.m. are plotted and colored for buffer control (black), Aβ and TDP-43 (red), Aβ (blue), and TDP-43 (green). Each group of fEPSPs before (black line) and after (red line) theta burst stimulation is shown individually in the upper panel. **a** For Aβ40 analysis, Aβ40 at 1 μM and TDP-43 at 10 nM were used. The slices were treated with buffer ($n = 4$ independent slices), Aβ40 with TDP-43 ($n = 3$, independent slices), Aβ40 alone ($n = 4$, independent slices), or TDP-43 alone ($n = 4$, independent slices). TDP-43 + Aβ40 vs. buffer; repeated two-way ANOVA, $p = 0.0032$, **$p < 0.01$. **b** For Aβ42 analysis, Aβ42 at 62.5 nM and TDP-43 at 0.625 nM were used. The slices were treated with buffer ($n = 9$ independent slices), Aβ42 with TDP-43 ($n = 4$, independent slices), Aβ42 alone ($n = 5$, independent slices), or TDP-43 alone ($n = 5$, independent slices). TDP-43 + Aβ42 vs. buffer; repeated two-way ANOVA, $p = 0.0232$, *$p < 0.05$. Source data are provided as a Source Data file.

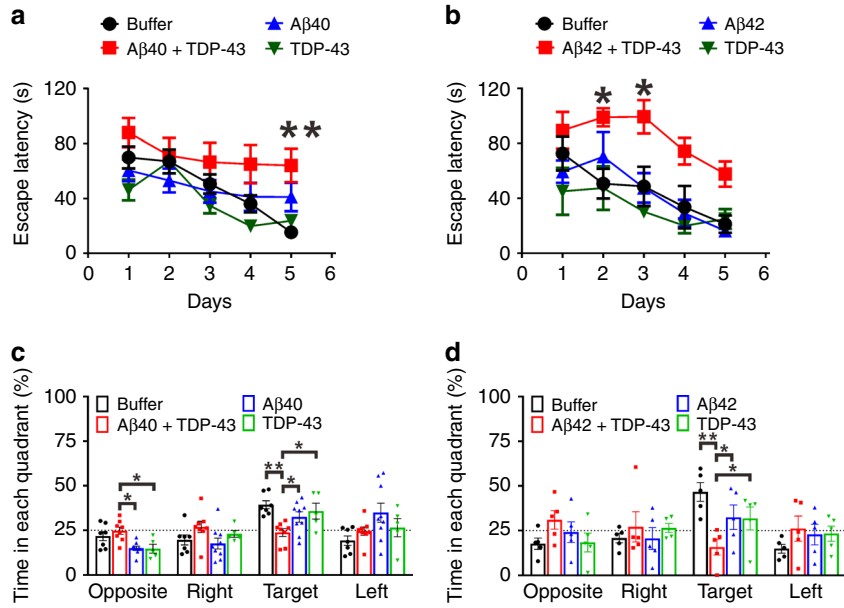

**Fig. 5 TDP-43-induced Aβ species impairs hippocampus-related spatial memory to a greater degree than Aβ or TDP-43 alone does in the injected mouse model.** WT mice at the age of 5-month were intracranially injected with the samples from the aggregation assays. Aβ, TDP-43, Aβ with TDP-43, and buffer obtained from the aggregation reaction were used. Spatial learning and memory functions were inspected by MWM 1 month after injection. Escape latency is defined as the time at which the hidden platform is found. The data were colored for buffer control (black), Aβ and TDP-43 (red), Aβ (blue), and TDP-43 (green). **a** Training phase for Aβ40. The sample sizes for buffer, Aβ40 with TDP-43, Aβ40, and TDP-43 are 7, 8, 9, and 5, respectively. The averaged data and s.e.m. are plotted. The statistical analysis was performed by repeated two-way ANOVA and Bonferroni's post-hoc test, **$p < 0.01$ (Buffer vs. Aβ40 + TDP-43 at day 5, $p = 0.0048$). **b** Training phase for Aβ42. The sample size for buffer, Aβ42 with TDP-43, Aβ42, and TDP-43 are 5, 4, 4, and 5, respectively. The averaged data and s.e.m. are plotted. The statistical analysis was performed by repeated two-way ANOVA and Bonferroni's post-hoc test, *$p < 0.05$ (Buffer vs. Aβ42 + TDP-43 at day 2, $p = 0.0174$; Buffer vs. Aβ42 + TDP-43 at day 3, $p = 0.0112$). **c** Probe test for Aβ40. The time in each quadrant were calculated in percentage. The averaged data and s.e.m. are plotted. The statistical analysis was performed by one-way ANOVA, Holm-Sidak's multiple comparisons, *$p < 0.05$, **$p < 0.01$ (In the opposite quadrant: Aβ40 + TDP-43 vs. Aβ40, $p = 0.0147$; Aβ40 + TDP-43 vs. TDP-43, $p = 0.038$; In the target quadrant, Buffer vs. Aβ40 + TDP-43, $p = 0.002$; Aβ40 + TDP-43 vs. Aβ40, $p = 0.0425$; Aβ40 + TDP-43 vs. TDP-43, $p = 0.0289$). **d** Probe test for Aβ42. The averaged data and s.e.m. are plotted. The statistical analysis was performed by one-way ANOVA, Holm-Sidak's multiple comparisons, *$p < 0.05$, **$p < 0.01$ (In the target quadrant: Buffer vs. Aβ42 + TDP-43, $p = 0.0019$; Aβ42 + TDP-43 vs. Aβ42, $p = 0.0311$; Aβ42 + TDP-43 vs. TDP-43, $p = 0.0311$). Source data are provided as a Source Data file.

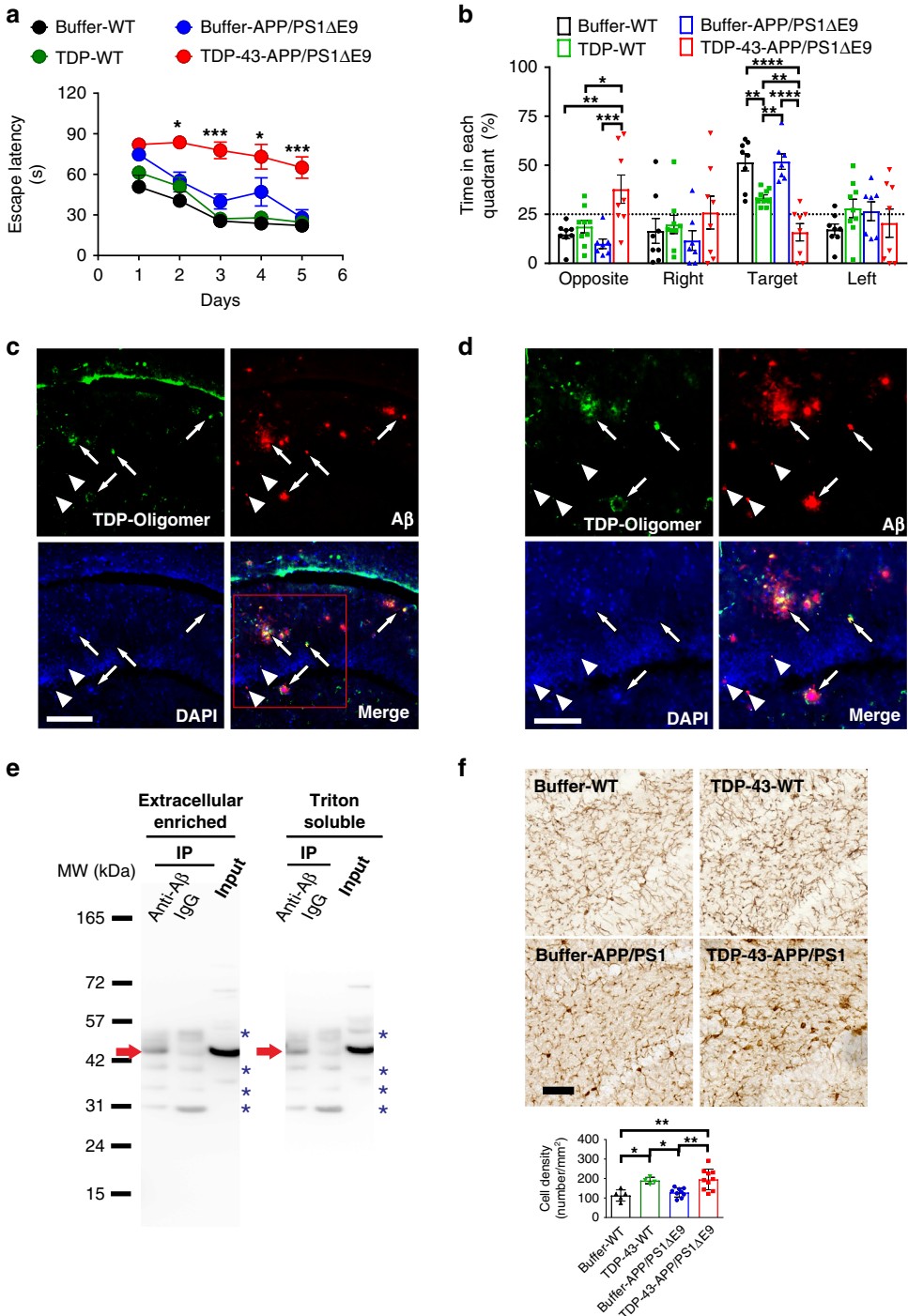

or without TDP-43 as described in Fig. 2. We first examined the dose-dependent toxicity of Aβ42 on LTP to find the concentration range for toxic Aβ (Supplementary Fig. 5, repeated two-way ANOVA, treatment factor: Buffer v.s. Aβ42-250 nM, $F = 50.21$, df 1/17, $p < 0.0001$; Buffer v.s. Aβ42-125 nM, $F = 10.95$, df 1/16, $p = 0.004$; Buffer v.s. Aβ42-62.5 nM, $F = 0.5199$, df 1/12, $p = 0.485$)). We found that Aβ42 at 250 and 125 nM were toxic, but not at 62.5 nM. To address TDP-43 effect, the samples were diluted to a non-toxic Aβ concentration at 62.5 nM to minimize Aβ42 toxicity. The same molar ratio of TDP-43 at 0.625 nM to Aβ were kept (1%). Similarly, TDP-43-induced Aβ42 significantly inhibited hippocampal LTP compared with that of the buffer control (repeated two-way ANOVA, treatment factor: $F = 7.454$, df 1/9, $p = 0.0232$) (Fig. 4b). No significant difference was found

in the fEPSP slope among the buffer control, 62.5 nM Aβ42, and 0.625 nM TDP-43 (repeated two-way ANOVA, treatment factor: $F = 0.5092$, df 2/15, $p = 0.611$). The result indicated that TDP-43-induced Aβ species significantly impair synapse function to a greater extent than Aβ alone or TDP-43 alone did.

Next, to investigate the detrimental effect of TDP-43-induced Aβ species on the hippocampus-related spatial learning and memory in vivo, we injected the aggregation samples to the hippocampi of wild-type mice and examined the mouse behavior by Morris water maze (MWM) paradigm 1 month after the injection (Fig. 5). Two microliters of the recombinant Aβ40 or 42 (25 μM), full-length TDP-43 (0.25 μM), TDP-43-induced Aβ (25 μM Aβ40 or 42, 0.25 μM TDP-43), or buffer control obtained from the aggregation assay were injected. In the pre-training

**Fig. 6 TDP-43 impairs spatial memory, colocalizes and interacts with Aβ, and increases microgliosis in APP/PS1ΔE9 mice. a, b** The spatial learning and memory function of APP/PS1ΔE9 mice were inspected via MWM. The training phase (**a**) and probe test (**b**) are shown (buffer, $n = 6$, TDP-43, $n = 8$). **a** For the training phase, the averaged data and s.e.m. are plotted. The data were colored for buffer-injected WT mice (black), TDP-43-injected WT mice (green), buffer-injected APP/PS1ΔE9 mice (blue), and TDP-43-injected APP/PS1ΔE9 mice (red). Statistical analysis was performed via repeated two-way ANOVA with Bonferroni's post-hoc test, *$p < 0.05$, ***$p < 0.001$ (Buffer-APP/PS1ΔE9 vs. TDP-43-APP/PS1ΔE9 at day 2, $p = 0.0190$; at day 3, $p = 0.0002$; at day 4, $p = 0.0135$; at day 5, $p = 0.0002$). **b** For the probe test, the average data and s.e.m. are plotted. Statistical analysis was conducted with one-way ANOVA, Holm-Sidak's multiple comparisons, *$p < 0.05$, **$p < 0.01$, ***$p < 0.001$, ****$p < 0.0001$ (In the opposite quadrant, Buffer-WT vs. TDP-43-APP/PS1ΔE9, $p = 0.0036$; TDP-WT vs. TDP-43-APP/PS1ΔE9, $p = 0.0122$; Buffer-APP/PS1ΔE9 vs. TDP-43-APP/PS1ΔE9, $p = 0.0008$; In the target quadrant: Buffer-WT vs. TDP-43-WT, $p = 0.0046$; Buffer-APP/PS1ΔE9 vs. TDP-43-APP/PS1ΔE9, $p < 0.0001$; TDP-WT vs. TDP-43-APP/PS1ΔE9, $p = 0.0046$; buffer-WT vs. TDP-43-APP/PS1ΔE9, $p < 0.0001$; TDP-WT vs. buffer-APP/PS1ΔE9, $p = 0.0046$). **c** Representative immunostaining micrographs in the hippocampus dentate gyrus show that TDP-43 oligomers colocalized with Aβ plaque (arrows) and intraneuronal Aβ (arrowhead; scale bar, 150 μm). Three induvial animals in each group were examined. **d** The enlarged view of the rectangle in 6c (scale bar, 100 μm). **e** Representative IP result of Aβ and TDP-43 in the brain fractions of APP/PS1ΔE9 mice injected with TDP-43. Extracellular-enriched and Triton-soluble brain fractions were used. IP was performed using Aβ antibodies and detected by TDP-43 antibodies. Immunoprecipitated TDP-43 was indicated by red arrows and nonspecific bands were indicated by blue asterisks. Four independent experiments were performed. **f** TDP-43 injection increased microgliosis in the hippocampus dentate gyrus of APP/PS1ΔE9 mice. Representative Iba1 immunostaining micrographs of buffer-injected ($n = 5$) or TDP-43-injected ($n = 4$) wild-type mice and buffer-injected ($n = 9$) or TDP-43-injected ($n = 10$) APP/PS1ΔE9 mice. The calculated cell density of Iba1-positive microglial cell is shown (scale bar, 30 μm). The averaged data and s.e.m. are plotted. Statistical analysis was performed by two-tailed Mann–Whitney test, *$p < 0.05$, **$p < 0.01$ (Buffer-WT vs. TDP-43-WT, $p = 0.0159$; Buffer-WT vs. TDP-43-APP/PS1ΔE9, $p = 0.0043$; TDP-43-WT vs. Buffer-APP/PS1ΔE9, $p = 0.0238$; Buffer-APP/PS1ΔE9 vs.TDP-43-APP/PS1ΔE9, $p = 0.0076$). Source data are provided as a Source Data file.

phase, all the groups showed a similar trend in a time-dependent decrease in escape latency demonstrating no sample injection effect (Supplementary Fig. 6a, b) (for Aβ40 study, repeated two-way ANOVA, time factor: $F = 54.47$; df 1/26; $p < 0.001$; treatment factor, $F = 2.875$; df 3/26; $p = 0.055$; for Aβ42 study, time factor: $F = 25.60$, df 1/15, $p = 0.0001$, treatment factor: $F = 0.82$, df 3/15, $p = 0.505$). The swimming velocity for each group was unchanged (Supplementary Fig. 6c, d) (for Aβ40 study, one-way ANOVA, $F = 0.016$, df 3/26, $p = 0.997$; for Aβ42 study, $F = 0.163$, df 3/16, $p = 0.354$).

In the training phase, the escape latency time of the TDP-43-induced Aβ40 (Fig. 5a) or Aβ42 (Fig. 5b) group was significantly increased compared with that of the buffer group (for Aβ40 study, repeated two-way ANOVA, $F = 5.25$, df 1/13, $p = 0.039$; for Aβ42 study, $F = 11.07$, df 1/8, $p = 0.010$). More importantly, the escape latency time of the TDP-43-induced Aβ was significantly increased compared with that of Aβ alone (for Aβ40 group, repeated two-way ANOVA, $F = 8.385$, df 1/14, $p = 0.012$; for Aβ42 group, $F = 19.14$, df 1/7, $p = 0.003$) and TDP-43 alone (for Aβ40 group, repeated two-way ANOVA, $F = 6.451$, df 1/11, $p = 0.0275$; for Aβ42 group, $F = 34.73$, df 1/8, $p = 0.0004$). In the probe test with the platform removed, the mice injected with TDP-43-induced Aβ40 (Fig. 5c) or Aβ42 (Fig. 5d) spent significantly less time than the other groups in the target quadrant (for Aβ40 study, one-way ANOVA, Holm-Sidak's multiple comparisons, Aβ + TDP vs. buffer, $p = 0.002$; Aβ + TDP vs. Aβ, $p = 0.0425$; Aβ + TDP vs. TDP-43, $p = 0.0289$; for Aβ42 study, Aβ + TDP vs. buffer, $p = 0.0019$, Aβ + TDP vs. Aβ, $p = 0.0311$, Aβ + TDP vs. TDP-43, $p = 0.0311$). The result indicated that TDP-43-induced Aβ significantly impaired spatial learning and memory of mice to a greater extent than Aβ or TDP-43 does.

**TDP-43 impairs spatial memory, increases amyloid plaques, interacts with Aβ, and induces microgliosis in APP/PS1ΔE9 mice.** To further investigate the effect of TDP-43 on AD-related pathology, we bilaterally injected TDP-43 protein into the hippocampus of APP/PS1ΔE9 mice and examined the resulting behavior and pathology. We first performed the MWM for 6-month-old APP/PS1ΔE9 mice and their wild-type littermates that received TDP-43 injection at the age of 5 months. In the pre-training phase, all the groups exhibited a similar time-dependent decrease in escape latency, demonstrating that the sample had no

injection effect (Supplementary Fig. 7a, repeated two-way ANOVA, time factor: $F = 35.26$; df 1/27; $p < 0.001$; treatment factor, $F = 1.964$; df 3/27; $p = 0.143$). The swim speeds did not differ among the groups (Supplementary Fig. 7b, one-way ANOVA, $F = 1.746$, df 3/27, $p = 0.181$). The results demonstrated that the visual ability and navigated motivation of all the groups were not impaired. In the training phase, the escape latency time of TDP-43-injected APP/PS1ΔE9 group was significantly increased compared with that of the buffer-injected group (Fig. 6a, repeated two-way ANOVA, time factor: $F = 8.291$, df 4/44, $p < 0.0001$; TDP-43 factor: $F = 24.03$; df 1/11; $p = 0.0005$). No difference was observed in the TDP-43-injected and buffer-injected wild-type mice (repeated two-way ANOVA, time factor: $F = 24.72$, df 4/60, $p < 0.0001$; TDP-43 factor: $F = 1.606$; df 1/15; $p = 0.2243$). In the probe test after the platform was removed, the mice in the TDP-43-injected APP/PS1ΔE9 or WT group spent significantly less time in the target quadrant than the buffer control group did. Furthermore, the memory function of the TDP-43-injected APP/PS1ΔE9 mice was more significantly impaired than that of the TDP-43-injected wild-type mice (Fig. 6b, one-way ANOVA, Holm-Sidak's multiple comparisons, buffer-WT vs. TDP-43-WT, $p = 0.0046$; buffer- APP/PS1ΔE9 vs. TDP-43-APP/PS1ΔE9, $p < 0.0001$; TDP-WT vs. TDP-43-APP/PS1ΔE9, $p = 0.0046$; buffer-WT vs. TDP-43-APP/PS1ΔE9, $p < 0.0001$; TDP-WT vs. buffer-APP/PS1ΔE9, $p = 0.0046$). The time spent of the TDP-43-injected APP/PS1ΔE9 in the opposite quadrant was longer than that of the other groups (one-way ANOVA, Holm-Sidak's multiple comparisons, buffer-WT vs. TDP-43-APP/PS1ΔE9, $p = 0.0036$, TDP-43-WT vs. TDP-43-APP/PS1ΔE9, $p = 0.012$, buffer-APP/PS1ΔE9 vs. TDP-43-APP/PS1ΔE9, $p = 0.0008$). Altogether, the results demonstrated that TDP-43 worsens the spatial memory in APP/PS1ΔE9 mice. In the pathological examination, we first analyzed the amyloid plaque burden in different brain areas. The Aβ level was quantified by immunostaining with Aβ antibody 4G8/6E10. The results showed that TDP-43 injection significantly increased the amyloid plaque burden in the olfactory bulb, the amygdala, the prefrontal cortex, the motor cortex, and the somatosensory cortex (Supplementary Fig. 8, repeated two-way ANOVA, TDP-43 factor: $F = 59.94$; df 1/56; $p < 0.0001$). We further analyzed TDP-43 oligomers and Aβ localization in the hippocampus dentate gyrus. We found that TDP-43 oligomers probed by poly TDP-O antibody were mostly intracellular and colocalized majorly with intraneuronal Aβ

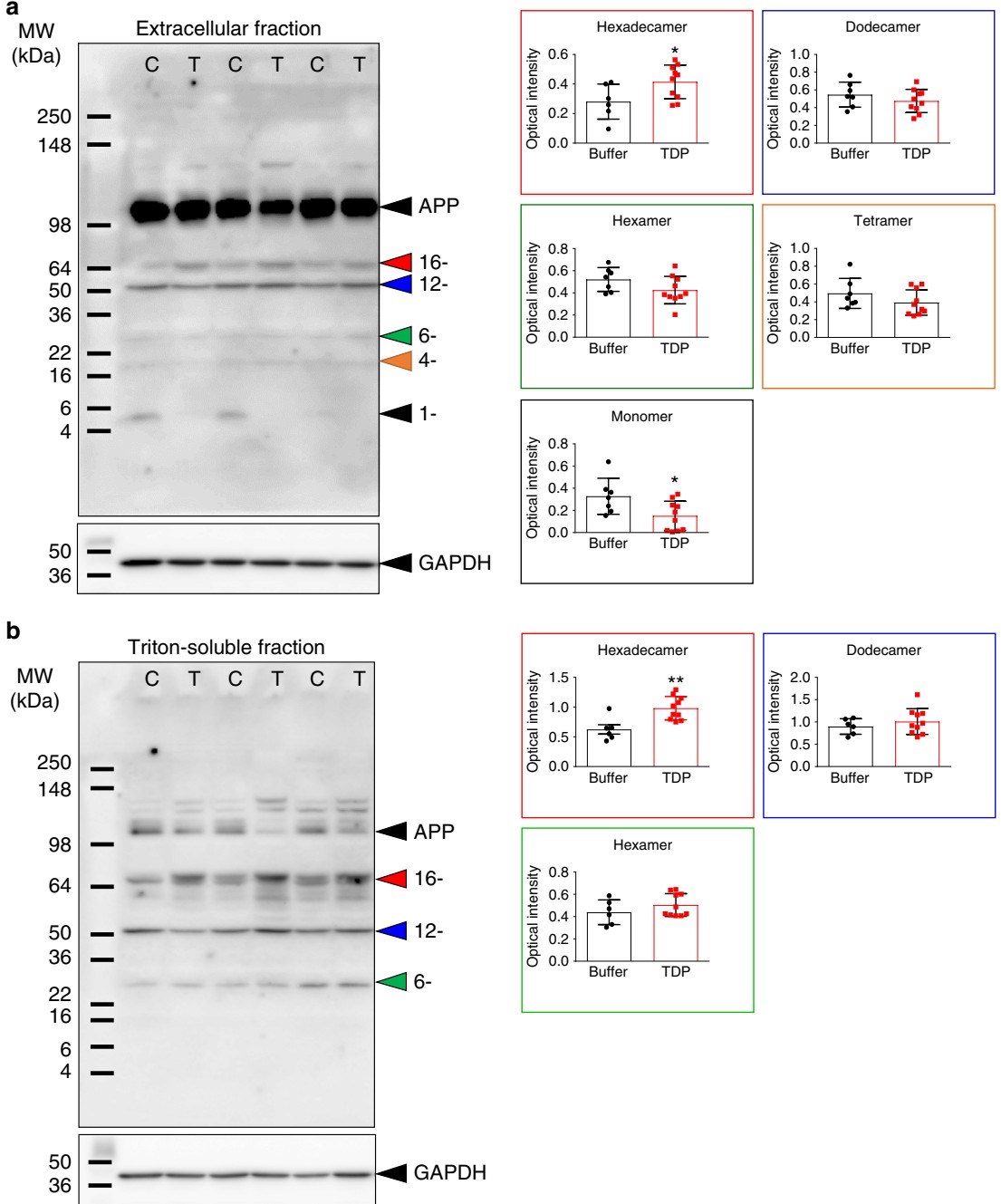

**Fig. 7 TDP-43 alters Aβ assembly in APP/PS1ΔE9 mice.** Representative western blot of **a** extracellular-enriched and **b** Triton-soluble fraction of TDP-43-injected or buffer-injected APP/PS1ΔE9 mice brain. Aβ was detected and quantified in different assembles. TDP-43 group is denoted as T, and the buffer group is denoted as C. The quantitative results of each Aβ assembly in TDP-43-injected ($n = 6$) or buffer-injected ($n = 10$) APP/PS1ΔE9 mice were shown on the right panels. The averaged data and s.e.m. are plotted. The statistical analysis was performed by two-tailed Mann–Whitney test, *$p < 0.05$, **$p < 0.01$ (for extracellular-enriched fraction, Hexadecamer: Buffer vs. TDP-43, $p = 0.042$; Monomer: Buffer vs. TDP-43, $p = 0.043$; For Triton-soluble fraction, Hexadecamer: Buffer vs. TDP-43, $p = 0.005$). Source data are provided as a Source Data file.

(~94%) and a few with amyloid plaques (~6%) (Fig. 6c, d). We further conducted immunoprecipitation (IP) to examine the interaction between Aβ and TDP-43 (Fig. 6e) in TDP-43-injected APP/PS1ΔE9 mice. We fractioned the mouse frontal cortex into extracellular-enriched fraction, Triton-soluble fraction, and guanidine hydrochloride (GdnHCl)-soluble fraction in accordance with the procedure described in the method. The mixture of Aβ antibodies 6E10 and 4G8 was used to pull down Aβ and the associated proteins in the extracellular-enriched and Triton-soluble fractions from the frontal cortex. The eluents were probed

with a mixture of two rabbit TDP-43 antibodies 10782-2-AP and 12892-1-AP. These antibodies recognize both human and mouse TDP-43. IgG was used as a negative control. The IP data showed that ~43 kDa TDP-43 band, most likely full-length TDP-43, was pulled down with Aβ but not with the negative control in the extracellular-enriched and Triton-soluble fractions. The result demonstrated that TDP-43 interacts with Aβ in extracellular-enriched and Triton-soluble fractions of the APP/PS1ΔE9 mouse frontal cortex. Furthermore, inflammation was examined by Iba1 staining for microgliosis. TDP-43 injection to APP/PS1ΔE9

mice resulted in increased microgliosis in the hippocampus dentate gyrus (Fig. 6f, Mann–Whitney test, Buffer-APP/PS1ΔE9 vs. TDP-43-APP/PS1ΔE9, $p = 0.008$, $U = 13$). Overall, these results showed that TDP-43 impairs spatial memory, increases amyloid burden, interacts with Aβ, and induces microgliosis in APP/PS1ΔE9 mice.

**TDP-43 alters Aβ assembly in APP/PS1ΔE9 mice**. Aβ assembly has been strongly implicated in AD pathogenesis[7,33]. Thus, we investigated whether the injected TDP-43 affects Aβ assembly. We employed western blot analysis to examine the Aβ assembly in different fractions from APP/PS1ΔE9 mouse brains with or without TDP-43 injection. In the extracellular-enriched fraction, we detected Aβ species, including monomer, tetramer, hexamer, dodecamer, and hexadecamer detected by a mixture of 6E10 and 4G8 antibodies. We found that the level of Aβ hexadecamer was significantly increased in the TDP-43-injected group (Fig. 7a, Mann–Whitney test, $p = 0.042$, $U = 11$), whereas, the Aβ monomer decreased (Mann–Whitney test, $p = 0.043$, $U = 14$). The levels of other Aβ assemblies were not changed (Mann–Whitney test, dodecamer: $p = 0.414$, $U = 26$; hexamer: $p = 0.088$, $U = 17$; tetramer: $p = 0.133$, $U = 19$). In the Triton-soluble fraction, we detected Aβ hexamer, dodecamer, and hexadecamer. Again, the level of Aβ hexadecamer was significantly increased in TDP-43-injected group (Fig. 7b, Mann–Whitney test, $p = 0.005$, $U = 5$). The levels of the other detectable Aβ assemblies were not changed (Mann–Whitney test, dodecamer: $p = 0.492$, $U = 23$; hexamer: $p = 0.300$, $U = 20$). We were unable to perform IP and western blot from the GdnHCl fractions because the high concentration of GdnHCl strongly affected the experiments. The result indicated that TDP-43 alters Aβ assembly in APP/PS1ΔE9 mice.

**TDP-43 oligomers are present and colocalize with intraneuronal Aβ in AD patients**. Previously, we found that TDP-43 oligomers are present in FTLD and ALS brains but not in age-matched controls. Above in vitro and in vivo studies showed that TDP-43 species, including TDP-43 oligomers interacted and colocalized with Aβ in the mouse brain. Therefore, to examine whether TDP-43 oligomers exist and colocalize with Aβ in AD patients, we applied immunostaining by poly TDP-O antibody or TDP-43 C-terminal antibody and Aβ antibody in the entorhinal cortex of four AD patients (3 male, 1 female) with a mean age of 75.5 ± 5.6 years old and a Braak stage of IV or V. In immunostaining analysis by poly TDP-O antibody (Fig. 8a), we found the signal colocalized mostly with intraneuronal Aβ (~68%) and little with amyloid plaques (~7%). About 25% of TDP-43 oligomer signal was not colocalized with Aβ. In immunostaining analysis by TDP-43 C-terminal antibody that recognized both native and misfolded TDP-43 (Fig. 8b), we found that about 38% of the TDP-43 signal colocalized with Aβ in which most signal colocalized with intraneuronal Aβ (~37%) and little with amyloid plaques (~1%). A large portion of the signals resided in the nucleus that is not colocalized with Aβ. We further used the neuronal marker MAP-2 to identify neurons via immunofluorescence. The result showed that Aβ and TDP-43 oligomers colocalized dominantly in the cytoplasm of neuronal cells (Fig. 9). The specificity of Aβ was also demonstrated by Aβ40 and Aβ42-specific antibodies (Supplementary Fig. 9a, b, respectively). We found that about 80% of the TDP-43 oligomer signal colocalized with Aβ40 and Aβ42 at a similar ratio. The TDP-43 oligomer signal mostly colocalized with intracellular Aβ (~78%) and slightly colocalized with amyloid plaques (~2%). No significant difference between Aβ40 and Aβ42 was observed. These results

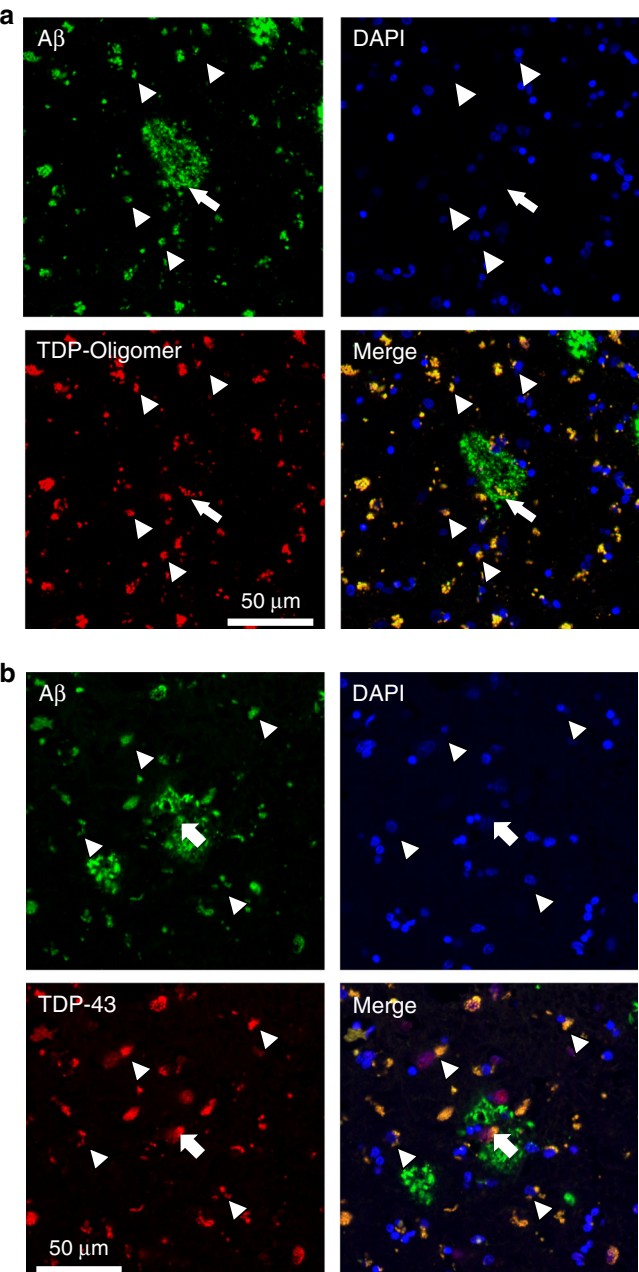

**Fig. 8 TDP-43 oligomers colocalize with amyloid plaques in the brain of an AD patient. a** Representative immunostaining micrographs reveal that TDP-43 oligomers mainly colocalized with intraneuronal Aβ (arrowhead) and partly with amyloid plaque (arrow). **b** Representative immunostaining micrographs show that both amyloid plaque (arrow) and intraneuronal Aβ (arrowhead) colocalized with the total TDP-43 in the entorhinal cortex of a 77-year-old patient with Braak stage IV AD. Four induvial samples were performed. Source data are provided as a Source Data file.

clearly showed that TDP-43 oligomers existed and colocalized mostly with intraneuronal Aβ in the brain of AD patients.

**More Aβ oligomers are present in AD patients with TDP-43 pathology**. We further used the hippocampal tissues of several cases with AD to examine their TDP-43 pathology as indicated by hyperphosphorylated TDP-43 (Supplementary Fig. 10b) in western blot and examined the level of Aβ assembly (Supplementary Fig. 10a). We found that the levels high-molecular-weight Aβ

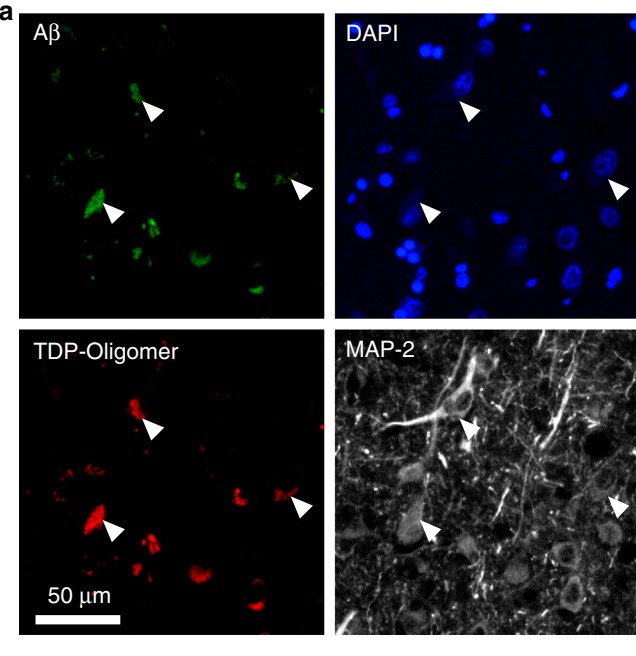

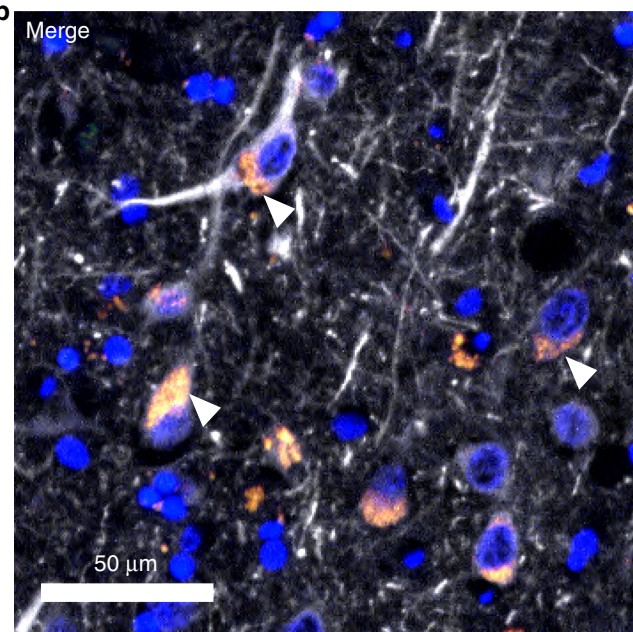

**Fig. 9 TDP-43 oligomers colocalize with Aβ in the cytoplasmic region of neurons of an AD patient.** Representative immunostaining micrographs reveal that intraneuronal Aβ (arrowhead) colocalize with TDP-43 in the entorhinal cortex of a 77-year-old patient with Braak stage IV AD. **a** Specimens are stained with Aβ antibodies (4G8 and 6E10), TDP-43 oligomer antibody, neuronal marker MAP-2, and DAPI. **b** The merged image is shown. The results demonstrated that many TDP-43 oligomers and Aβ colocalized in the cytoplasm of neuronal cells. Four induvial samples were performed. Source data are provided as a Source Data file.

species of AD patients with hyperphosphorylated TDP-43 ($n = 3$) were lower and their levels of Aβ oligomers were higher than those of AD patients but without TDP-43 pathology ($n = 5$). Aβ oligomer at 50 kDa was significantly higher in AD patients with TDP-43 pathology than in AD patients but without TDP-43 pathology. These results were consistent with those observed in APP/PS1ΔE9 mice injected with TDP-43.

## Discussion

Aβ progressively aggregates into amyloid fibril via a nucleation-elongation mechanism. In this study, we found that TDP-43 oligomers affected the conformational change in Aβ and prevented the formation of Aβ fibrils. This condition may be attributed to the amyloidogenic properties of TDP-43 since a similar phenomenon has also been observed in other amyloidogenic protein-protein interaction[34,35]. For example, cystatin C (CysC) forms soluble amyloid oligomers that inhibit Aβ aggregation more potently than monomeric and dimeric CysC do[34]. Transthyretin tetramer suppresses Aβ aggregation and ameliorates AD phenotype by interacting with Aβ[35]. TDP-43 possibly formed a complex with Aβ and caused detrimental effects although only 1% TDP-43 was present in the Aβ samples under our experimental conditions. However, we encountered a technical difficulty in separating a defined aggregated species and maintaining the original property because TDP-43 and Aβ both form oligomers. We further found that the inhibition of Aβ fibrillization by TDP-43 only occurs in the early and intermediate stages of Aβ fibrillization, but not in the fibrillar stage and the seeding reaction. The result demonstrated that TDP-43 does not affect mature Aβ fibrils and cannot block fibril growth once the fibril nucleus was added. It also suggests that TDP-43 cannot block secondary nucleation in Aβ fibrillization. For the discrepancy found in the inhibitory effect of TDP-43 on Aβ40 and Aβ42, Aβ42 may not be effectively inhibited by TDP-43 because Aβ42 initially forms several oligomers with few monomers and rapidly fibrillizes[36–38]. Furthermore, Aβ40 and Aβ42 oligomerize through distinct pathways[36–38]. Freshly prepared Aβ40 predominantly forms monomers[38], forms a monomer-added mixture after crosslinking[36], and slowly fibrillizes[6,38,39]. By contrast, freshly prepared Aβ42 forms paranuclei[36,37] or tetramers[37,38] in equilibrium with monomer, and rapidly aggregates. Therefore, the potential interaction site and species of Aβ40 and Aβ42 for TDP-43 can vary.

In the characterization of the specificity and interaction of TDP-43 and Aβ, we found full-length TDP-43, TDP-43_265, and TDP-43_N-term, not TDP-43_RRM 1 + 2, potently inhibited Aβ fibrillization; however, TDP-43_RRM1 + 2 has strong interaction with Aβ. Since full-length TDP-43 formed oligomers and TDP-43_265 and N-term formed dimeric species, the result might indicate that oligomerization/dimerization is required for the inhibitory activity, where the interaction may be majorly attributed from RRM1 + 2 and minorly, if any, from N-terminal region. Therefore, we hypothesized that TDP-43 mainly interacted with Aβ through RRM domains and required dimeric/oligomeric TDP-43 to inhibit Aβ fibrillization. This inhibition resulted in the retention of Aβ oligomers. TDP-43_265 was likely present because of the calpain cleavage of TDP-43 in the brain of AD patients, considering that calpain is implicated in AD[40] and calpain-cleaved TDP-43 was found in the brain and spinal cord of ALS patients[41].

In the animal studies, we demonstrated that the modification of Aβ aggregation by TDP-43 increased neurotoxicity by impairment of synapse transmission in LTP reduction ex vivo and spatial memory in wild-type mice in vivo. In TDP-43-injected APP/PS1ΔE9 mice, TDP-43 impaired spatial memory and increases Aβ-related pathologies, including amyloid plaque burden, Aβ assembly, and microgliosis. The result was consistent with previous studies showing that TDP-43 overexpression induces neuroinflammation in a lentiviral gene transfer rat model[24] and that depletion of TDP-43 in APP/PS1ΔE9 mice decreases the amyloid plaque burden in hippocampus and cortex[42]. Although we found TDP-43 inhibits Aβ fibrilization in vitro, we suggest that brain inflammation induced by TDP-43 injection leads to an increase of the amyloid plaques in vivo. Our

studies showed that TDP-43 plays multiple roles in AD; for example, TDP-43 induces more Aβ oligomers by directly interacting with Aβ, leading to synaptic dysfunction and memory impairment; TDP-43 increases brain inflammation that worsens spatial memory and causes an increase in amyloid burden[43,44]. TDP-43 also affects APP metabolism[45]. These events could occur in the same periods, but they affected AD pathology to different degrees.

We showed that TDP-43 oligomers colocalized largely with intraneuronal Aβ and some with amyloid plaques in APP/PS1ΔE9 mice and AD brain. TDP-43 interacted with Aβ in both the extracellular-enriched and Triton-solution fractions of APP/PS1ΔE9 mouse brain after TDP-43 injection. We suggest that soluble Aβ oligomers in the extracellular-enriched and Triton-solution fractions interacted with TDP-43 since amyloid plaques were retained in GdnHCl fractions. Intracellular Aβ has been observed in both sporadic and familial AD patients[46,47] and was reported to be an early event in AD and in Down syndrome preceding extracellular Aβ[47,48]. The interaction site for TDP-43 and Aβ in cell could be in cytoplasm and/or lysosome. Aβ and TDP-43 have been reported to influence endosomal-lysosomal pathways[49,50] where both aggregates can be cleared by autophagy mechanism[51,52]. Extracellular Aβ can be re-internalized and found inside endosomal/lysosomal compartments[53]. Cytosolic Aβ can also be found without vesicle structure[54]. Passive leakage along the pathways and active uptake of extracellular Aβ by several cell surface receptors, leading to intracellular accumulation have been reported[9]. Intracellular Aβ has been reported to bind to several cytosolic chaperons like Hsp70 and small heat shock proteins[55], cytosolic enzymes such as SOD1[56], and directly interact with amyloid binding alcohol dehydrogenase in mitochondria[57].

Furthermore, in our TDP-43-injected APP/PS1ΔE9 mice, a specific Aβ assembly hexadecamer is significantly increased. Previous studies showed that injection of Aβ dodecamers impairs the memory of young rats[33], and Aβ dimers isolated from AD brains impair synaptic plasticity and memory[58]. These studies have highlighted the importance of specific oligomeric Aβ for memory impairment. Thus, we propose that the spatial memory impairment in TDP-43-injected APP/PS1ΔE9 mice is due to increased inflammation and accumulation of Aβ oligomers, which is considered more toxic than Aβ fibrils and monomers[7]. However, although we showed that TDP-43 induced both Aβ40 and Aβ42 toxicity in LTP and in water maze study in the injection mice model and demonstrated that both Aβ40 and Aβ42 colocalized in brain of post mortem AD patients, the exact molecular interaction of TDP-43 to Aβ40 and Aβ42 needs to be further investigated to elucidate potential differences resided in Aβ40 and Aβ42. Overall, our results demonstrated that TDP-43 inhibited Aβ fibrillization by interacting with Aβ and exacerbated AD-related pathology.

## Methods

**TDP-43 cloning, expression, and purification.** For TDP-43 proteins, the plasmids used for expression are TDP-43_FL, a pET14b vector containing cDNA of full-length TDP-43, TDP-43_265 (a pET14b vector containing the cDNA of TDP-43 aa. 1–265), TDP-43_N-term (a pQE30 vector containing the cDNA of TDP-43 aa. 1–101), and TDP-43_RRM1 + 2 (a pQE30 vector containing the cDNA of TDP-43 aa. 101-265). Full-length TDP-43 was cloned previously[22]. TDP-43_265 was cloned by deletion mutagenesis of full-length TDP-43 with forward primer and reverse primer listed in Supplementary Table 1. TDP-43_N-term and TDP-43_RRM1 + 2 are gifts from Dr. Hanna S. Yuan, Institute of Molecular Biology, Academia Sinica[59]. The pET14b plasmids were transformed into *E. coli* strain Rosetta 2 (Novagen, Merck KGaA), and the pQE30 plasmids were transformed into *E. coli* strain M15. All TDP-43 proteins contain His-tag in the N-terminal region. Full-length TDP-43 and TDP-43_265 contain extra N-terminal residues MGSSHHHHHHSSGLVPRGSHMLE. TDP-43_N-term and TDP-43_RRM1 + RRM2 contained extra N-terminal residues MRGSHHHHHHGS. *E. coli* were

grown in LB medium with 50 μg/ml ampicillin at 37 °C to OD600 around 0.4 and induced by 0.5 mM IPTG at 16 °C. After 22 h, the cells were harvested and lysed in 30 mM Tris-HCl buffer, pH 8, containing 500 mM NaCl, 10% glycerol, 1 mM dithiothreitol (DTT), 2% RNase A, 2% DNase I, and protease inhibitor cocktail (Complete, EDTA-free, Roche Applied Science, Mannheim, Germany). The supernatant was collected after centrifugation and loaded onto a Ni- NTA affinity column (GE Healthcare Bio-Sciences AB, Uppsala, Sweden). The column was first equilibrated using a buffer containing 30 mM Tris, pH 8, 500 mM NaCl, 1 mM DTT, 20 mM imidazole, and 10% glycerol. Proteins were eluted with a step gradient of imidazole in the same running buffer. The purity of His-tagged TDP-43 protein was checked on SDS–PAGE and identified by Coomassie blue staining. Proteins were stored at −20 °C freezer and further dialyzed with 10 mM Tris at pH 8 right before the experiments. The dialyzed protein was centrifuged at 17,000 × g at 4 °C for 30 min to remove all precipitates. Protein concentration was quantified by micro BCA protein assay kit (Thermo Fisher Scientific, USA).

**Aβ preparation.** Recombinant Aβ expression and purification were performed following our previous study[60]. No additional amino acid was generated. The purified Aβ peptide was lyophilized and stored at −80 °C. Biotinylated Aβ was purchased from Biopeptide Inc. (San Diego, CA, USA).

**ThT assay.** For ThT assay shown in Fig. 1a, Aβ40 powder, 0.1 mg, was first treated with hexafluoroisopropanol (HFIP), lyophilized, then dissolved in 100 μL, 3 mM NaOH and lyophilized. The lyophilized powder was then dissolved in 100 μL, 10 mM Tris buffer, pH 7.8, and Aβ stock concentration was determined via BCA assay. The Aβ alone sample was prepared from a stock solution by adding more Tris buffer to reach 25 μM. As the concentration of freshly dialyzed TDP-43_FL solution is very low (~0.25 μM quantified by Micro BCA assay), TDP-43_FL in 10 mM Tris buffer, pH 7.8 was directly added into another 0.1 mg Aβ powder to avoid the decrease of concentration of TDP-43. This condition did not contain any organic solvent or denaturant. The samples were placed in Eppendorf tubes and shaken for 20 s at 500 rpm every h in Thermomixer C (Eppendorf, Germany) at 25 °C. The selected time point samples were collected for ThT measurement at 25 °C.

For ThT assay shown in Fig. 1e–h and Fig. 2, Aβ40 peptide at 0.1 mg was treated with HFIP and lyophilized for at least 18 h before use. Aβ40 sample was dissolved in 6 μl dimethyl sulfoxide (DMSO) and diluted into 94 μl, 10 mM Tris buffer at pH 8. The concentration of Aβ stock was determined by BCA protein assay kit (Thermo Fisher Scientific). An aliquot of freshly prepared Aβ stock was diluted into TDP-43 samples or buffer containing 5 μM ThT to prepare Aβ solution at 25 μM. The final DMSO concentration in the samples is less than 2%. To determine the effect of TDP-43 protein on Aβ oligomers or on Aβ fibrils, the Aβ stock was first incubated for 20 h at 4 °C with continuous shaking at 300 rpm to prepare Aβ oligomers or with continuous shaking at 1000 rpm for 24 h to prepare Aβ fibrils. The Aβ species were validated in TEM. TDP- 43 and control buffer were added to Aβ oligomer or fibrils and subjected to ThT assay. For seeding experiment, the 2.5 μM Aβ fibrils solution was added at the beginning of aggregation assay of Aβ at 25 μM. The samples were incubated in a 384-well opaque microtiter plate, sealed with a transparent film, and monitored by a microplate reader (SpectraMax M5, Molecule Devices) at 25 °C. The ThT signals were measured every hour with a 60-s mixing before the measurement. ThT fluorescence was excited at 442 nm, and the emission spectrum was collected at 485 nm by software SoftMax Pro 6.3 in SpectraMax M5. Data were plotted and analyzed in GraphPad Prism 7.0.

**Dot blotting.** To probe the conformational change of Aβ40 during fibrillization, 2 μL of Aβ samples with and without TDP-43 were collected from different time points from the ThT study shown Fig. 1a. They were dotted onto nitrocellulose membranes and probed by anti-amyloid oligomers A11 polyclonal antibody (1:1000, Life Technologies) and anti-amyloid fibrils OC polyclonal antibody (1:10,000, Merck Millipore). For full-length TDP-43, freshly dialyzed TDP-43_FL and variants were prepared at the aforementioned condition and the concentration was determined via micro BCA assay. Two microliters of TDP- 43 variants at 0.25 μM were dotted onto nitrocellulose membranes. After blocking with 5% skim milk in TBST, the membrane was blotted with polyclonal TDP-43 oligomer-specific antibody, poly TDP-O[1] (1:4000 in blocking solution) and anti-rabbit horse-radish peroxidase (HRP)-conjugated secondary antibodies (1:5000, Merck Millipore).

**Circular dichroism.** The time-course samples collected for CD measurement were prepared in the same condition as for the ThT assay in Fig. 1a. All far-ultraviolet CD spectra were recorded by Spectra Manager 2.0 in Jasco J-815 spectropolarimeter (Jasco Inc.) using a circular quartz cell (Hellma) with 1 mm path length at room temperature (RT). Each spectrum was collected from 250 to 190 nm, corrected with buffer background, and averaged from 10 scans. Data were plotted in GraphPad Prism 7.0.

**Transmission electron microscopy.** Five microliters of TDP-43 proteins or Aβ samples from ThT assay were deposited onto 400-mesh Formvar carbon-coated copper grids (EMS electron Microscopy Sciences, Hatfield, PA, USA) for 1 min.

The grids were blotted, washed with droplets of Milli-Q water, and stained with 2% uranyl acetate. The samples were examined with a FEI Tecnai G2 F20 Super TWIN transmission electron microscope with an accelerating voltage of 120 kV. For immunogold labeling, 5 μl Aβ fibrils with TDP-43 was placed on grids for 5 min, then washed, blocked, and probed following previous protocol[22] but with primary antibody using anti-TDP-43 N-term rabbit antibody (10782, 1:1000, Proteintech). In the end, the grids were incubated with an 18 nm gold-conjugated secondary anti-rabbit IgG antibody (1:40, Jackson ImmunoResearch) at RT for 1 h. The unbound antibody was heavily washed by high-salt Tween and PBS. The grids were then fixed by 1% glutaraldehyde in PBS at RT for 10 min and washed six times by double-distilled H₂O. Finally, the grids were negatively stained using 2% uranyl acetate.

**Enzyme-linked immunosorbent assay**. The concentration of freshly purified TDP-43_FL and truncation proteins were quantified by Bradford protein assay (Thermo Fisher Scientific) and the same mole number of each variant at 1 μM (200 pmole in 200 μL) was immobilized in 96-well ELISA microplates (Thermo Fisher Scientific, USA) by incubation overnight at 4 °C. HFIP and NaOH-treated biotin-labeled Aβ40 was prepared in 10 mM Tris at pH 7.8 to make a series of Aβ solution with concentrations ranging from 0 to 38 μM. After immobilization, the plate was blocked for 2 h at RT with 5% skim milk in TBST (50 mM Tris-Cl, pH 7.6, 150 mM NaCl, 0.1% Tween-20). The plates were washed and probed by biotin-Aβ40 solution for 2 h at RT. After wash, the plates were subjected to HRP-conjugated Streptavidin (1:5,000, Merck Millipore) for 1 h at RT. After another wash, the color was developed by adding 100 μL 3,3,5,5 -tetramethyl benzidine (Merck Millipore). The reaction was stopped with 100 μL 250 mM HCl and the absorbance was recorded at 450 nm by software SoftMax Pro 6.3 in SpectraMax M5 (Molecular Device). Data were plotted and analyzed in GraphPad Prism 7.0.

**Biolayer interferometry analysis**. HFIP- and NaOH-treated N-terminally biotinylated Aβ40 was dissolved in 10 mM Tris, pH 7.8 with 0.005% Tween-20 to make a concentration of 10 μM solution. The sensors used are streptavidin-coated (SA sensor, ForteBio). The sample sensors and reference sensors were also pre-incubated in the same buffer before loading. The biotinylated Aβ40 was immobilized to sensors and then the sensors were subjected to TDP-43_FL or TDP-43 variants solution to probe the interaction between Aβ and TDP-43 proteins. In general, all steps were performed at 37 °C with continuous agitation at 1,000 rpm. The association and dissociation time were both 300 s. The regeneration buffer was 10 mM glycine at pH 1.5. The sensorgrams were measured with double references (reference buffer and reference sensor) using an Octet Red96 and $K_D$ value was obtained by fitting the data using the Data Analysis Software (ForteBio).

**SDS-PAGE and western blot for proteins**. The ThT assays were stopped by adding SDS-PAGE sample buffer and then the samples without heating were loaded onto a 13.3% Tris/tricine separating gel with 10% and 4% stacking portions. Separated Aβ species on the gel were electrophoretically transferred to a nitro-cellulose membrane at 250 mA for 75 min at 4 °C. The membrane was blocked with 5% skim milk in TBST. Aβ samples were probed by 6E10 antibody (1:5000, Bio-Legend) and anti-mouse IgG secondary antibody. The signals were visualized by ECL detection kit (Merck Millipore).

**Animal**. All experiments were done in accordance with the National Institutes of Health Guideline for Animal Research (Guide for the Care and Use of Laboratory Animals) and Taiwan Animal Protection Law and were approved by the Academia Sinica Institutional Animal Care and Utilization Committee (IUCAC 16-02-939). C57BL/6 mice were obtained from the National Laboratory Animal Center (Taipei, Taiwan) and the APP/PS1ΔE9 (B6C3-Tg(APPswe,PSEN1dE9)85Dbo/Mmjax) mice were obtained from the Jackson Laboratory (Bar Harbor, ME, USA). All of the APP/PS1ΔE9 mice were genotyped using a protocol provided by the Jackson Laboratory. All of the mice were housed (4–5 per cage) at a stable temperature (23 ± 1 °C), humidity 55 ± 5%, and unrestricted access to food and water with a light cycle from 7 am to 7 pm during the experimental period. We used 3-4 mice per group for the electrophysiology study, 5-9 C57/BL6JNarl mice per group and 6–8 APP/PS1ΔE9 mice per group for the Morris Water Maze (MWM), 3 APP/PS1ΔE9 mice per group for the immunostaining staining, 9–10 APP/PS1ΔE9 mice per group for microgliosis staining, and 6–10 APP/PS1ΔE9 mice per group were used for Aβ western blotting.

**Electrophysiological analysis**. Hippocampal slice for field excitatory postsynaptic potential (fEPSP) was prepared as previously described[61]. The 12-week-old C57BL/6JNarl male mice were anaesthetized by isoflurane and decapitated, then the mice brain was quickly removed and placed into the ice-cold artificial cerebrospinal fluid (119 mM NaCl, 2.5 mM KCl, 1.3 mM MgSO₄, 2.5 mM CaCl₂, 26.2 mM NaHCO₃, 11 mM Glucose and1.25 mM NaH₂PO₄, aCSF). The hippocampus was removed and then cut into 450 μm thickness by vibratome (Leica, Nussloch, Germany). The hippocampal slices were recovered for 2 h at room temperature in a chamber with aCSF, which bubbled with a mixture of 95% O2 and 5% CO₂. For extracellular recordings, the hippocampal slices were placed at the center of a MED–P515A/5 (1 mm) probe (Alpha MED Scientific Inc., Osaka, Japan) with 64 embedded

recording electrodes containing with circulating aCSF and 100 μM picrotoxin bubbled with a mixture of 95% O₂ and 5% CO₂. The fEPSP was recorded from the Schaffer collaterals fibers on the stratum radiatum layer of hippocampal CA1 area by using MED64 multichannel recording system in Neuro- electrophysiology core, Academia Sinica. Test stimuli were given every 30 s (0.033 Hz), and the stimulus intensity was set to produce 40% of the maximum spike-free response. A stable baseline was recorded for at least 30 min before treatment with aCSF containing Aβ, full-length TDP-43, or TDP-43-induced Aβ at indicated concentrations. The samples were prepared following Aβ preparation for ThT assay for Fig. 2[32] and collected at ~50 h of incubation. After treatment with protein or respective buffer control for 30 min, LTP was induced by theta burst stimulation (10 bursts of 4 stimuli at 100 Hz, with an inter-burst interval of 200 ms) given at baseline intensity according to our previous publication[61]. The fEPSP slope was measured from ~10–90% of the rising phase using a least squares regression. The data were quantified by normalization to the baseline slope and mean ± SEM are shown. Repeated two-way ANOVA was used for statistical analysis. The respective buffer control was performed on the same mice but different brain slice to exclude individual bias.

**Intracranial injection to mice**. Five-month-old male C57BL/6JNarl mice or APP/PS1ΔE9 male mice were anaesthetized by intraperitoneal injection of a mixture of tranquilizer (Zoleti 50, 1 mg/10 mg body weight, Vibrac, Amherst, MA, USA), analgesics (Rompun, 10 μL/10 mg body weight, Bayer, Toronto, Canada). The C57BL/6JNarl mice received bilateral intrahippocampal injection of 2 μL recombinant TDP-43 or Aβ42 (25 μM), full-length TDP-43 (0.25 μM), or TDP-43-induced Aβ (25 μM Aβ40 or Aβ42 with 0.25 μM TDP- 43). APP/PS1ΔE9 mice received bilateral intrahippocampal injection of 2 μL recombinant full-length TDP-43 (2.5 μM). The stereotaxic coordinates of the injection site are in relation to bregma as follows: Posterior 2 mm; mediolateral 1 mm; ventral 2 mm. The injection needle was slowly approached to the desired depth and the mentioned protein was injected using a microsyringe (0.1 μL/min, 32-gauge Hamilton Company, NV, USA). The needle left in place for an additional 5 min to limit the diffusion of the injected protein.

**Morris water maze test**. The MWM was performed in a custom-made circular pool with a diameter of 154 cm and a wall height of 60 cm, which was filled with opaque water (diluted milk) at a temperature of 20 ± 2 °C and depth of 32 cm. During the pre-training trials, the circular escape platform made of transparent Plexiglas (diameter 13 cm) was emerged 0.3 cm above the water surface and kept location constant for 2 days. In the training trial, the escape platform was submerged 0.5 cm below the water surface and the location of the hidden platform was moved to another place and kept constant for five days. Animals were given a session training (1,000–1,800 h) per day during pre- and training trails. Each session consisted of four swim trials (120 s per trial) with different quadrant starting positions for each trial. After the last day of training, the mice were given a probe test. During the probe test, animals were placed in the center position of the pool, and the mice were allowed to swim for 60 s without the platform presented. The whole process was recorded by a charge-coupled device camera and the escape latency (i.e., time to reach the platform, in seconds), time spending in target quadrant of probe test, path length and swim speed (cm/s) were analyzed by EthoVision video tracking system (Noldus Information Technology, Wageningen, Netherlands).

**Brain tissue preparation and fractionation**. The mice were anesthetized by intraperitoneal injection of a mixture of tranquilizer (Zoleti 50, 1 mg/10 mg body weight, Vibrac, Amherst, MA, USA), analgesics (Rompun, 10 μL/10 mg body weight, Bayer, Toronto, Canada) and perfused from the left ventricle with ice-cold PBS, and their brains were quickly removed. The left hemispheres were stored in −80 °C immediately. The right hemispheres were fixed in 10% formalin at 4 °C for frozen section and subjected to immunostaining. The brain was cut into an average of 130 pieces of 30-μm coronal sections. For analysis of amyloid plaque burden or TDP-43 oligomer signals, the brain sections were measured in every 12th section of the brain from anterior (stereotaxic reference: bregma 2.5 ± 0.2 mm) to posterior part (stereotaxic reference: bregma −6.5 ± 0.2 mm).

The left hemispheres were collected, homogenized, and subjected to different fractionations following previous literature[33]. For extracellular-enriched protein fraction, 25% (w/v) lysis buffer containing 50 mM Tris-HCl (pH 7.6), 0.01% NP-40, 150 mM NaCl, 2 mM EDTA, 0.1% SDS, protease and phosphatase inhibitor (VWR Life Science Products, Philadelphia, USA) was used to homogenize the frozen brain tissues with a 1 ml syringe, gauge 19 needle. The homogenates were centrifuged for 10 min at 1,000 × g and the supernatant was collected. The pellet was washed by repeating the homogenizing step again. For Triton-soluble fraction, 25% (w/v) lysis buffer containing 50 mM Tris-HCl (pH 7.6), 150 mM NaCl, 1% Triton X-100 (Merck, Darmstadt, Germany) was added to the pellet from the previous centrifugation and pipetted to homogenize the pellet. The homogenates were further centrifuged for 90 min at 16,000 × g and the supernatant was collected. The pellet was washed by repeating the homogenizing step again. For guanidine HCl soluble fraction, 25% (w/v) lysis buffer containing 5 M guanidine HCl (GdnHCl) in 50 mM Tris-HCl (EMD-Millipore, Darmstadt, Germany) was added

to the pellet obtained after centrifugation of the Triton-soluble fraction and pipetted to homogenize the pellet.

**Immunostaining**. The detailed procedure for immunohistochemistry was descri- bed in previous studies[62,63]. In short, coronal sections (30 μm thickness) were mounted on poly-L lysine coated slides (Thermo Fisher Scientific, Waltham, MA, USA). The coronal section was antigen retrieved by citric acid buffer (10 mM citric acid, pH 6.0, 0.05% Tween-20) at 80 °C for 30 min, then blocked with 3% BSA in PBS containing 0.5% Triton X-100 for 1 h. For immunofluorescence, the coronal section was probed with the primary antibodies: a mixture of 6E10 (1:2000, SIG- 39320, BioLegend, San Diego, CA, USA) and 4G8 (1:2000, SIG-39220, BioLegend, San Diego, CA, USA) or polyclonal TDP- 43 oligomer antibody, poly TDP-O[22] (1:1000). The secondary antibody used for 4G8 and 6E10 is goat anti-mouse Alexa Fluor 488 (A28175, 1:1000, Invitrogen, Camarillo, CA, USA) and for poly TDP-O is goat anti-rabbit Alexa Fluor 594 (A-11012, 1:1000, Invitrogen, Camarillo, CA, USA). Fluorescence images were taken by Aperio FL Digital Pathology Scanner (Leica Biosystem, Mannheim, Germany). The signals were measured by ImageJ (NIH, Bethesda, MD, USA) and a given background threshold was subtracted. The background intensity threshold was fixed and applied to all sections. For immu- nohistochemistry, rabbit anti-ionized calcium-binding adaptor molecule 1 (Iba1, 1:1000, 091-19741, Wako, Japan) antibody was used. The Iba1 signal was then developed using an immunohistochemistry kit (Super Sensitive Polymer-HRP IHC Detection System, Biogenex, Fremont, CA, USA) and the secondary antibody (1:1,000) provided in the kit. The 3, 3´-diaminobenzidine (DAB) was used as the substrate to develop immunohistochemistry signal. To analyze protein expression level, the slide images were taken by Aperio AT2 Digital Pathology Scanner (Leica Biosystem, Mannheim, Germany). All of the quantitative signals were obtained by calculating the signal intensities within the area that were higher than a back- ground threshold with the aid of ImageJ 1.8.0_60 (NIH, USA). To prevent the injection injury effect, the brain slice from the injection side anterior to posterior 0.5 mm would not be used for quantification. The intensity threshold of back- ground was fixed and applied to all images.

**Co-Immunoprecipitation (IP)**. Co-IP was performed to examine the interaction between Aβ and TDP-43. The buffer and TDP-43 injected APP/PS1ΔE9 mice brain samples were taken for analysis as described in brain tissue preparation and fractionation. The extracellular-enriched, Triton-soluble, and GdnHCl fractions were taken for BCA assay to measure the total protein concentration. Two hundred micrograms of the sample was used for co-IP and 100 μg for the input. The fractions were diluted with filtered TBS-T to a total volume of 200 μl. Two microliters of mouse monoclonal 4G8 antibody against Aβ aa. 17–24 (SIG-39220, BioLegend, San Diego, CA, USA) and 2 μl of mouse monoclonal 6E10 antibody against Aβ aa. 1–17 (SIG-39320, BioLegend, San Diego, CA, USA) was added to the samples for IP study. Mouse IgG2a (mabg2a-ctrlm, InvioGen, San Diego, Cali- fornia, USA) was served as IgG control for this experiment. The samples with the antibodies were mixed overnight in a vertical rotator (F1–10 program, ELMI, Intelli Mixer RM-2L). On the second day, 10 μl of Protein A Mag (Sepharose Xtra, 1 ml, GE healthcare, Life sciences, Missisauga, Canada) was added to the sample and mixed in the vertical rotator for about 4 h in 4 °C. Then, the samples were washed with TBS-T for three times in a magnetic rack (20-400. Magna GrIP rack, EMD Millipore, Darmstadt, Germany) and eluted with a 6x loading buffer (25% glycerol, 50 mM Tris-base, 4% SDS, and 0.04% β- mercaptoethanol), then heated for 10 min at 95 °C. Next, the samples were subjected to western blot with gradient gels (NuPAGE 4–12% Bis-Tris Gel 1.0 mm*15 well, Invitrogen, Carlsbad, CA) in mini gel tanks (A25977, Thermo Fisher Scientific, Waltham, MA, USA) using MES running buffer (B0002, 20x Bolt, MES SDS running buffer, Thermo Fisher Sci- entific, Waltham, MA, USA) and transferred to PVDF membrane (Amersham Hybond P 0.45, GE healthcare, Chicago, Illinois, USA). The membrane was blocked with 5% non-fat milk and the TDP-43 antibodies, a mixture of a rabbit polyclonal TDP-43 antibody (1:1000, 10782-2-AP, Proteintech, Rosemount, USA) and TDP-43 C-terminal antibody (1:1000, 12892-1-AP, Proteintech, Rosemount, USA), were applied for detection. The images were obtained by Imagequant LAS4000 system (GE Healthcare, Life sciences, Hungary).

**Western blot for Aβ in mice**. Western blot was performed to observe the changes in Aβ assembly after injecting TDP-43 to the APP/PS1ΔE9 mice. The concentra- tion of the extracellular-enriched fraction and Triton-soluble fraction were deter- mined and adjusted to the same protein concentration (48 μg in 20 μL). The samples were mixed with 6x loading buffer and heated in 95 °C for 10 min, then loaded into a 4–15% Tris-glycine gel and resolved at 100 V for 120 min. The separated proteins were transferred to a PVDF membrane (Amersham Hybond P 0.45, GE healthcare, Chicago, Illinois, USA) and blocked with 5% non-fat milk. The mouse monoclonal 6E10 antibody (1:5000, SIG-39320, BioLegend, San Diego, CA, USA) and mouse monoclonal 4G8 antibody (1:5000, SIG-39220, BioLegend, San Diego, CA, USA) were mixed and used for Aβ detection. A mouse monoclonal GAPDH antibody (1:5000, Proteintech, Rosemont, IL, USA) was used for loading control. The secondary antibody used for 6E10/4G8 mixture is goat anti-mouse IgG-HRP (GTX213111-01, Genetex, CA, USA) and for GAPDH is goat anti-rabbit IgG-HRP (GTX213110-01, Genetex, CA, USA). Immobilon Western

Chemiluminescent HRP substrate (Merck, Darmstadt, Germany) was used for developing. The detection was carried out with Imagequant LAS4000 system (GE Healthcare, Life sciences, Hungary) and analyzed using ImageJ (NIH, Bethesda, MD, USA).

**Human brain staining**. All of the human tissue-related procedures and usage were approved by the Human Subjects Research Ethics, Academia Sinica, Taiwan. Participants were recruited and consented in the Alzheimer's Disease Center at University of California, Davis, Sacramento, California. To investigate the inter- action of the TDP-43 oligomer and Aβ in Alzheimer's disease patient, the 5 μm- thickness paraffin-embedded sections were deparaffinized with xylene. To remove xylene and rehydrate the slides, the specimens were washed through graded ethanol in water and ending with ddH₂O. Then, the specimens were immersed in citric acid buffer (10 mM citric acid, pH 6.0, 0.05% Tween-20) and heated for antigen retrieval at 80 °C for 30 min, and then blocked with 3% BSA in PBS with 0.5% Triton X-100 for 1 h. To observe the amyloid plaque and TDP-43, TDP-43 C-terminal antibody (1:500, 12892-1-AP, Proteintech, Rosemount, USA), poly TDP-O[1] (1:1000), Aβ antibody (6E10 (1:2000, SIG-39320, BioLegend, San Diego, CA, USA)/ 4G8 (1:2000, SIG-39220, BioLegend, San Diego, CA, USA)), Aβ40-specific antibody 11A50 (1:500, SIG-39140, BioLegend, San Diego, CA, USA), Aβ42-specific antibody 12F4 (1:500, SIG-39142, BioLegend, San Diego, CA, USA) and neuronal marker MAP-2 (1:500, 188004, Synaptic systems, Goettingen, Saxony, Germany) were applied to the slides for 2 h in room temperature. After that, the primary antibodies were washed out by PBS/0.5% Triton X-100 and probed with the secondary antibodies for 2 h in room temperature. The secondary antibodies used were goat anti-rabbit Alexa Fluor 594 (A-11012, 1:1000, Invitro- gen, Camarillo, CA, USA) for TDP-43 C-terminal antibody and poly TDP-O, goat anti-mouse Alexa Fluor 488 (A28175, 1:1000, Invitrogen, Camarillo, CA, USA) for Aβ, and goat anti-guinea pig Alexa Fluor 647 (A-21450, 1:1000, Invitrogen, Camarillo, CA, USA) for MAP-2. Subsequently, the nuclei were stained with Hoechst Stain solution (1:1000, 0.5 ng/ml, H6024, Sigma-Aldrich, St. Louis, Mis- souri, USA). Lastly, glycerol gelatin (GG1-10X15ML, Sigma-Aldrich, St. Louis, MO, USA) was applied to mount each tissue slides. After the staining procedure, the human brain slide images were captured by Aperio CS2 Digital Pathology Scanner (Leica Biosystem).

**Reporting summary**. Further information on research design is available in the Nature Research Reporting Summary linked to this article.

## Data availability
Data for figures of this manuscript and supplementary information are available in the source data file. Other data from the corresponding authors upon request. Source data are provided with this paper.

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

## Acknowledgements

We thank research founding from Career Development Grant, Academia Sinica (AS-CDA-106-L01), Genomics Research Center, Academia Sinica, and Ministry of Science and Technology, Taiwan (MOST 108-2113-M-001-027). The work was supported in part by the U.S. National Institute on Aging grant P30 AG10129 (University of California Davis Alzheimer's Disease Research Center). We thank Dr. Hanna S. Yuan, Institute of Molecular Biology, Academia Sinica for providing pQE30 plasmids and *E. coli* strain M15. We thank Dr. Shu-Chuan Jao of the Biophysics Core Facility, funded by Academia Sinica Core Facility and Innovative Instrument Project (AS-CFII108-111), and Mr. Hong-Chang Chu in Dr. Ying-Da Wu's laboratory at Genomics Research Center, Academia Sinica for providing technical assistance of biolayer interferometry experiments.

We thank TEM Core Facility, Academia Sinica, for assisting TEM imaging. We thank National Laboratory Animal Center (NLAC), NARLabs, Taiwan, for technical support in contract breeding and testing services and Dr. Sin-Jhong Cheng of the Neuro-Electrophysiology Core, Neuroscience Program of Academia Sinica (NPAS) for providing technical support in LTP studies.

## Author contributions

L.H.T. designed the biochemical studies; Y.H.S. designed the animal studies; L.H.T., T.Y.C., and Y.T.L. performed the biochemical studies; Y.H.S. and K.G. performed the animal studies; Y.H.S., K.G., and P.S.C. performed the immunostaining of human brain tissues; W.W.C. performed western blot of human brain tissues. Y.S.F cloned the plasmid TDP-43 1–265; L.W.J. provided clinical assessment and human brain tissues; Y.H.S., L.H.T., K.G., W.W.C., T.Y.C., and Y.R.C. analyzed the data and wrote the manuscript; and Y.R.C. conducted the research direction.

## Competing interests

The authors declare no competing interests.
