## [Peer Review File · Nature Communications]

Reviewers' comments:

Reviewer #1 (Remarks to the Author):

Shih et al. described about the role of TDP-43 on amyloid-beta (A β) fibrillization in vitro and in vivo, and investigated the colocalization of TDP-43 with A β in the postmortem brains of patients with Alzheimer disease (AD), as well. The authors reported that TDP-43 inhibited A β fibrillization at initial and oligomeric stages, and impaired spatial memory in the wild-type mice and exacerbated AD-like pathology in APP/PS1deltaE9 mice when full-length TDP-43 was injected into the bilateral hippocampi. They also showed the colocalization of TDP-43 oligomer immunoreactivity with A β immunoreactivity in the neuronal cytoplasm in the autopsied brains of patients with AD.

As the significance of this paper is promotion of our understanding as to why AD patients with TDP-43 pathology exhibit more severe memory loss and hippocampal atrophy compared with those without TDP-43 pathology, authors should more clearly describe what is the mechanism underlying the exacerbation of AD phenotype in AD with TDP-43 pathology: i.e. If the mechanism is an increase of A β monomer/oligomers in the presence of TDP-43 resulting from the inhibition of fibrillization of A β , demonstration of an increase of A β monomer/oligomers in the brains of AD patients with TDP-43 pathology compared to those without TDP-43 pathology is necessary, and so on.

Although TDP-43 bound to both intracellular and extracellular A β in IP experiment (Fig. 6d), authors claimed that TDP-43 was found in association with A β monomer/oligomers but not with fibrillized A β and that the majority of extracellular amyloid plaques were negative for TDP-43. As TDP-43 strongly bound to A β (Table 1), readers would like to know how TDP-43 binds to A β in the cytoplasm and dissociates from it after A β is secreted extracellularly.

N-terminal fragments of TDP-43, including TDP_265 used in this experiment (Fig. 2), are produced in the ALS/FTD brains by activated calpain-dependent cleavage (Yamashita et al., Nature Commun. 3:1307, 2012). TDP_265 had more potent inhibitory effects against A β fibrillization than full-length TDP-43, suggesting that calpain-dependent TDP-43 N-terminal fragments may play more significant roles than full-length TDP-43 in the AD pathogenesis. Therefore, demonstration of their presence in AD brains may facilitate our understanding of the AD pathogenesis.

Specific comments:

Line 218 : Application of TDP-43 variants should be in the mole-base, as the differences in the MW causes difference of interacting molecule among the TDP-43 variants.

Line 316: According to M&M, 10782-2 antibody is not TDP-43 C-terminal-specific.

Line 403: The authors described that TDP-43 oligomers colocalized largely with intraneuronal A β and some with extracellular amyloid plaques in AD and AD model mouse brains. Fig. 6c is too small to confirm their claim. Fig. 6c should be replaced or added with figures of larger magnification, and authors should indicate colocalization of TDP-43 with intracellular and extracellular A β more clearly. Fig. 6d indicates that TDP-43 seems to be co-precipitated with A β in the samples from both extracellular and Triton-X-soluble fractions, suggesting TDP-43 is bound to both intracellular and extracellular A β . The band for TDP-43 precipitated with A β antibody in the extracellular fraction appears to be more dense than those in the Triton-soluble fraction. The IP results seem to be contradictory to their claim that TDP-43 is found to be associated with intracellular A β rather than extracellular amyloid plaques.

Fig. 1: No fibrillization kinetics data on A β 42. As A β 42 was handled same as A β 40, kinetics data were necessary. Legend: A β appeared after Line 606 should be A β 40.

Fig. 2c: Results are indicated two points at 100h and 150h, which are not comparable with Fig. 1c,d, where results are from the >160h point. Fig. 2d : A β 40 aggregates at the gel top were more abundant in the samples of _265 than full-length TDP-43, whereas ThT intensity was higher in full-length than _265 in Fig. 2b. Explanation for the discrepancy is needed.

Fig. 4: Explain the black and red lines in the sample recording data.

Fig.6: As the behavioral changes of the APP/PS1deltaE9 mice were observed after 12 months of age, do the untreated mice behave normally at 6 months of age in this experiment? Fig. 6c: Brain area should be indicated. There are only 3 arrows in the panel of A β instead of 4 in other panels. Fig. 6d: The band of the sample 4G8+6E10 is quite different from the input. What denotes the band in the IgG-precipitated samples? Is there non-specific binding between TDP-43 and immunoglobulin? WB figure illustrating entire SDS-PAGE should be shown as TDP-43 fragments are usually found in the brain and increase in pathological conditions. Fig. 6e: brain areas should be indicated. If the hippocampi are the area, distance from the injection site should be also indicated. Legend: (f) should appear after "and" in Line 677.

Fig. 8/ Fig. S7 : As TDP-O Ab recognizes not only oligomers but also monomers (Ref. 18), the IR does not necessarily indicate only oligomers. Colocalization of TDP-43 with A β should be illustrated more clearly in figures using e.g. cytoplasmic markers of neurons.

Fig. S6 : Amyloid burden is not clear in the figures. Figures with higher contrast are required. Injection site should be indicated. As amyloid is detected in the olfactory tract (Line 304), figures at the level of the olfactory bulb are necessary.

As there are more than one APP/PS1 mouse line, notation of a more accurate manner as APP/PS1deltaE9 or APP/PS1 Δ E9 is better.

Reviewer #2 (Remarks to the Author):

This is an interesting manuscript in which the authors have investigated the relationship between TDP-43 and amyloid-beta. The authors have found that TDP-43 inhibits amyloid-beta fibrillization at initial and oligomeric stages. Amyloid-beta fibrillization was delayed specifically in the presence of N-terminal domain containing TDP-43 40 variants, while C-terminal TDP-43 was not essential for amyloid-beta interaction. The authors also found that TDP-43 significantly enhances amyloid-beta ability to impair long-term potentiation and causes spatial memory deficit. These findings were consistent with results from experiments on Alzheimer's disease (AD) transgenic mice injected with TDP-43 primed amyloid-beta. Despite the topic of the manuscript is significant and might unravel important aspects of AD pathology, this reviewer has several concerns that reduce enthusiasm towards the overall manuscript. Specifically, the reviewer has the following criticisms:

a) Why have the authors performed aggregation experiments in vitro using Amyloid-beta 40 instead of oligomer prone amyloid-beta 42? One would have liked to see the results of the same experiments with the 42aa form of the peptide, as one cannot exclude that the longer peptide produces different results. Moreover, in the second part of the manuscript the authors used amyloid-beta 42 primed preparations of unknown characterization.

b) This reviewer is not clear on the fate of TDP-43 in the preparations in which Amyloid-beta is primed with it. Has it been removed? Is it possible that the combination of amyloid-beta plus TDP-43 is responsible for the observed effects, instead of the modified amyloid-beta alone? Moreover, would it be possible that amyloid-beta and TDP-43 form mixed oligomers that are responsible for the observed effects?

c) It is surprising to this reviewer that high concentrations of amyloid-beta (especially for the

experiments performed with the 42aa peptide) alone do not impair LTP and behavior. One wonders the type of amyloid-beta used for the experiments shown on the manuscript. It does not seem that the recombinant amyloid-beta undergoes oligomerization through a method inspired by the classical method by Stine et al (J Biol Chem. 2003 Mar 28;278(13):11612-22)

d) All the experiments on APP/PS1 mice lack controls in non-transgenic mice treated with vehicle and those treated with amyloid-beta and/or TDP-43. These controls should have been performed in interleaved experiments with those on transgenic mice.

Minor criticisms:

- a) The manuscript does not specify the concentration used for amyloid-beta 40 and 42 in the behavioral experiments shown on figure 5. The text reports a concentration (25 uM) for amyloid-beta without defining the length of the peptide. Did the authors use the same concentration for amyloid-beta 40 and 42? Why, considering that amyloid-beta 42 is much more toxic than amyloid-beta 40? This reviewer is worried that with these high concentrations the authors are interfering with mechanisms other than those related to synaptic dysfunction (i.e. cell death).
- b) One would have liked to have displayed findings from all quadrants in the probe trial of the Morris water maze test, to make it sure that the assay is properly performed.

Reviewer #3 (Remarks to the Author):

In the present paper by Shih et al. the authors studied the effect of TDP-43 on Amyloid-beta aggregation and have extended earlier work which showed that full-length TDP-43 forms oligomers and agitates fibrillization (Fang YS Nat Communications, 2014). The main finding of the present study is that TDP-43 inhibited and delayed A β fibrillization and oligomerization. Moreover, TDP-43 injection into an AD mouse model (APPPS1) increased A β plaque load and impaired spatial memory in those mice. The authors propose that TDP-43 inhibits A β fibrillization but exacerbates AD pathology. These findings are somehow contradictory. As detailed below, the authors over-interpret their findings and there are some technical concerns. I have specific comments that the authors can tackle in a straightforward revision.

Specific comments:

- 1) The in vitro aggregation data of Figure 1 suggests, that there is a prolonged lag phase in the presence of TDP-43. Furthermore the authors show that Ab fibrillization is reduced by TDP-43. On the other hand the authors show that amyloid plaque load was significantly increased in several brain regions upon TDG-43 injection into APPPS1 transgenic mice. These data are contradictory and need to be discussed in detail. It is not enough to mention in the discussion section on page 12 that 'The discrepancy among these studies indicates.....is complex.....and should all be considered'.
- 2) Although the second sentence in the discussion implies that 'TDP-43 oligomers affect a conformational change in A β thereby preventing the formation of Ab fibrils' – this has not been shown and it is an open question on how this should work (mechanism).
- 3) It is not clear why TDP-43 impairs spatial memory? Is this due to more A β plaques? Please clarify and explain.
- 4) Additional parameters should be used to control for the animals behavior in the pool such as test for visual ability and motivation (e.g. swim speed).
- 5) The study would improve if the in vitro data would be repeated with rigorous in vivo seeding experiments in young pre-depositing APPPS1 transgenic mice that should be analyzed at different time points (time course). Due to the prolonged lag phase in vitro one would expect reduced Ab seeding. Those data should be more comparable to the in vitro data.
- 6) Although the authors claim in the last sentence of the discussion that TDP-43 exacerbates neuronal damage of A β and AD related pathology they did not study or show neuronal damage at all. Therefore this sentence has to be at least rephrased.

Here, we have performed several new experiments and added new data to answer the reviewers' questions. The point-by-point responses to the reviewers' comments are listed below.

Reviewers' comments:

Reviewer #1 (Remarks to the Author):

Shih et al. described about the role of TDP-43 on amyloid-beta (A β) fibrillization in vitro and in vivo, and investigated the colocalization of TDP-43 with A β in the postmortem brains of patients with Alzheimer disease (AD), as well. The authors reported that TDP-43 inhibited A β fibrillization at initial and oligomeric stages, and impaired spatial memory in the wild-type mice and exacerbated AD-like pathology in APP/PS1deltaE9 mice when full-length TDP-43 was injected into the bilateral hippocampi. They also showed the colocalization of TDP-43 oligomer immunoreactivity with A β immunoreactivity in the neuronal cytoplasm in the autopsied brains of patients with AD.

Q1: As the significance of this paper is promotion of our understanding as to why AD patients with TDP-43 pathology exhibit more severe memory loss and hippocampal atrophy compared with those without TDP-43 pathology, authors should more clearly describe what is the mechanism underlying the exacerbation of AD phenotype in AD with TDP-43 pathology: i.e. If the mechanism is an increase of A β monomer/oligomers in the presence of TDP-43 resulting from the inhibition of fibrillization of A β , demonstration of an increase of A β monomer/oligomers in the brains of AD patients with TDP-43 pathology compared to those without TDP-43 pathology is necessary, and so on.

A1: We thank the reviewer for this comment. To evaluate the situation in AD patients is a difficult task since each individual is different in genetics, age, and race. At this point, we only have a few frozen tissues that allow us to investigate A β assembly by Western blot. In this revised manuscript, we compared hippocampi from AD subjects with TDP-43 pathology (indicated by hyperphosphorylated TDP-43, n=3) with those from AD subjects without TDP-43 pathology (n=5) (see Supplementary Figure 10).

We found that the AD patients carrying TDP-43 pathology have a trend to harbor less A β fibrils (High-Molecular-Weight species stocked on top of the gel) and more A β oligomers (indicated by yellow arrowheads). Among them, the amount of ~50 kDa A β oligomers is significantly higher in AD with TDP-43 pathology than AD without TDP-43 pathology. The trend supports the presence of TDP-43 pathology in AD increases A β oligomers and decreases A β aggregation. We have added the data in the result (Supplementary Figure 10) and discussed the mechanism in the discussion in our revised manuscript.

Q2: Although TDP-43 bound to both intracellular and extracellular A β in IP experiment (Fig. 6d), authors claimed that TDP-43 was found in association with A β monomer/oligomers but not with fibrillized A β and that the majority of extracellular amyloid plaques were negative for TDP-43. As TDP-43 strongly bound to A β (Table 1), readers would like to know how TDP-43 binds to A β in the cytoplasm and dissociates from it after A β is secreted extracellularly.

A2: We thank the reviewer for this comment. Although the colocalization of TDP-43 with extracellular plaque is fewer, there is interaction with soluble A β in both extracellular-enriched and intracellular fractions. We explained the discrepancy between IP and IF results in more details in the answer to Q6. Our result is consistent with the in vitro protein study that TDP-43 interacts with A β monomer/oligomers more than fibrils.

The reviewer's suggestion for us to address how TDP-43 binds to A β in the cytoplasm and dissociates from it after A β is secreted extracellularly is excellent. However, the detailed dynamics could be complicated and difficult to monitor in live cells. To address this question requires substantial efforts of carefully designed full investigations with specialized cellular dynamic techniques and a good cellular tracking system. We hope we can build the system and investigate this question in the near future.

Q3: N-terminal fragments of TDP-43, including TDP_265 used in this experiment (Fig. 2), are produced in the ALS/FTD brains by activated calpain-dependent cleavage (Yamashita et al., Nature Commun. 3:1307, 2012). TDP_265 had more potent inhibitory effects against A β fibrillization than full-length TDP-43, suggesting that calpain-dependent TDP-43 N-terminal fragments may play more significant roles than full-length TDP-43 in the AD pathogenesis. Therefore, demonstration of their presence in AD brains may facilitate our understanding of the AD pathogenesis.

A3: We thank the reviewer for this comment. Here, we examined the level of calpain 1 and 2 as well as TDP-43 in frozen AD brain tissues. We did not see difference in the level of calpain (see below figure, panel A) and TDP-43 (see below figure, panel B). We used N-terminal TDP-43 antibody to probe TDP-43 and saw two bands of TDP-43, the full-length and the 35 kDa, presumably ~1-265. However, the levels of the two TDP-43 bands were similar. Therefore, at this point we cannot further comment on this question. The purpose of using TDP-43 1-265 is to compare to other variants according to their structural domains for protein-protein interaction. Therefore, we did not include the following data, but have mentioned about the possible presence of calpain-cleaved TDP in AD in the discussion.

Specific comments:

Q4: Line 218 : Application of TDP-43 variants should be in the mole-base, as the differences in the MW causes difference of interacting molecule among the TDP-43 variants.

A4: We thank the reviewer to point out the mistake. In the revised manuscript, we have repeated this experiment using same mole number, 200 pmole, for each TDP-43 variants to coat on ELISA plate (see Fig. 3a) and revised the description in the methods in SI. The data were consistent with our previous result.

Q5: Line316: According to M&M, 10782-2 antibody is not TDP-43 C-terminal-

specific.

A5: We are sorry for the mistake. Only 12892-1-AP is a C-terminal-specific TDP-43 antibody. Therefore, we have changed the description as “The eluents were probed with a mixture of two rabbit TDP-43 antibodies 10782-2-AP and 12892-1-AP.”

Q6: Line 403: The authors described that TDP-43 oligomers colocalized largely with intraneuronal A β and some with extracellular amyloid plaques in AD and AD model mouse brains. Fig. 6c is too small to confirm their claim. Fig. 6c should be replaced or added with figures of larger magnification, and authors should indicate colocalization of TDP-43 with intracellular and extracellular A β more clearly.

Fig. 6d indicates that TDP-43 seems to be co-precipitated with A β in the samples from both extracellular and Triton-X-soluble fractions, suggesting TDP-43 is bound to both intracellular and extracellular A β . The band for TDP-43 precipitated with A β antibody in the extracellular fraction appears to be more dense than those in the Triton-soluble fraction. The IP results seem to be contradictory to their claim that TDP-43 is found to be associated with intracellular A β rather than extracellular amyloid plaques.

A6: We thank the reviewer for pointing out the confusions. In the revised manuscript, we have added a zoom-in image for Fig 6c (see Fig 6d). The colocalization of TDP-43 oligomers with A β plaque (arrows) and intraneuronal A β (arrowheads) is indicated. For the question about the discrepancy between immunofluorescence images and IP result, we think it is due to the procedure of different methods. For the brain tissue fractioning, we first used a buffer to dissolve extracellular proteins to obtain the extracellular-enriched fraction according to previous literature (Lesne, et al. 2006). In this reference, they have also shown various A β oligomers in the extracellular-enriched fraction. A β in this fraction should be more soluble, whereas, A β plaques should still remain in the insoluble part and later dissolved in the GdnHCl fraction. This could explain the differences observed in IP and in the colocalization study for TDP-43 and A β . Therefore, we suggest soluble A β interacts with TDP-43 oligomers in both extracellular space and intracellular space.

Q7: Fig. 1: No fibrillization kinetics data on A β 42. As A β 42 was handled same as A β 40, kinetics data were necessary. Legend: A β appeared after Line 606 should be

A β 40.

A7: We thank the reviewer for this comment. In our first submission, we have shown the inhibitory effect of TDP-43 variants on A β 42 fibrillization in Supporting Figure 4. We have explained the reason in text as below.

“We also tested the inhibitory effect of TDP-43 variants on A β 42 fibrillization by ThT assay. The ThT result showed that full-length TDP-43 still significantly inhibited A β 42 fibrillization by retarding the lag time from 20 h to 40 h (**Supplementary Fig. 4**). However, the other TDP-43 variants did not significantly affect A β 42 fibrillization which may be due to rapid aggregation of A β 42 that reduces the effective interaction between TDP-43 and A β 42.”

Line 606: A β in figure legend 1 has been revised to A β 40.

Q8: Fig. 2c: Results are indicated two points at 100h and 150h, which are not comparable with Fig. 1c,d, where results are from the >160h point.

A8: We are sorry for the confusion. The sample preparation and incubation condition for Fig 1a-1d and Fig 2 were different. For Fig 1, we aim to study conformational changes for each time point by ThT assay and far-UV CD (data added in this revision). Since DMSO cannot be used in CD measurement, we prepared A β without DMSO and the samples were placed in Eppendorf tubes. The samples were shaken for 20 sec at 500 rpm every hour in Thermomixer C (Eppendorf, Germany). We collected the samples at indicated time points and performed ThT and CD measurement. For Fig 2, A β was prepared with DMSO and the protein samples were automatically measured by a microplate reader (SpectraMax M5, Molecule Devices) every hour and with a 60-sec mixing prior to the measurement. We stopped the plate reader specifically at 100 h to obtain the sample and continue incubation to 150 h to collect another sample for TEM (Fig. 2c). Hence, the discrepancy found in TEM should be due to DMSO treatment and the incubation condition that have been shown to affect A β fibrillization kinetics. All detailed preparations were described in the methods in SI.

Q9: Fig. 2d : A β 40 aggregates at the gel top were more abundant in the samples of _265 than full-length TDP-43, whereas ThT intensity was higher in full-length than _265 in Fig. 2b. Explanation for the discrepancy is needed.

A9: We thank the reviewer for this comment. The samples in Fig 2d were collected at the end of the aggregation after 150 hr and have been frozen for several days before Western blot, whereas, ThT assay is a real-time measurement and TEM samples are freshly dotted and air dried for later measurement. We suspect the discrepancy is due to the frozen process that affected the samples. Since the aggregates on top of the gel is not very quantitative which may depend on the epitope exposure of the aggregates, we focused on analysis of the lower portion of the SDA-PAGE. We have described this point more clearly in the text.

Q10: Fig. 4: Explain the black and red lines in the sample recording data.

A10: We thank the reviewer for this comment. The black line is fEPSPs before the theta burst stimulation and the red line is after. We have added the explanation in the figure legend for Fig 4.

Q11: (1) Fig.6: As the behavioral changes of the APP/PS1 Δ E9 mice were observed after 12 months of age, do the untreated mice behave normally at 6 months of age in this experiment ? **(2)** Fig. 6c: Brain area should be indicated. There are only 3 arrows in the panel of A β instead of 4 in other panels. **(3)** Fig. 6d: The band of the sample 4G8+6E10 is quite different from the input. What denotes the band in the IgG-precipitated samples? **(4)** Is there non-specific binding between TDP-43 and immunoglobulin? **(5)** WB figure illustrating entire SDS-PAGE should be shown as TDP-43 fragments are usually found in the brain and increase in pathological conditions. **(6)** Fig. 6e: brain areas should be indicated. If the hippocampi are the area, distance from the injection site should be also indicated. Legend: (f) should appear after “and” in Line 677.

A11: We thank the reviewer for the questions. The questions are addressed point-by-point as follows:

- (1) We have added the wild type littermate control in this experiment, at 6 months of age the buffer treated WT and buffer treated APP/PS1 Δ E9 mice showed no differences in hippocampus related spatial learning and memory (Fig. 6a and 6b).
- (2) Sorry for missing the information. The colocalization was analyzed in the hippocampus dentate gyrus. We have added the information in text and figure legend 6c. The problem of the arrows is fixed.
- (3)–(5) The shifted band in the original image in Fig 6d is due to a bad gel. We

replaced the image with a better one and showed the whole blot. The major band immunoprecipitated is full-length TDP-43 indicated by a red arrow. Non-specific bands were indicated.

(6) We have indicated the brain area the hippocampus dentate gyrus in text and in figure legend (now Fig 6f). The injection site has been described previously in method “**Intracranial injection to mice**: The stereotaxic coordinates of the injection site are in relation to bregma as follows: Posterior 2 mm; mediolateral 1 mm; ventral 2 mm..”. To prevent the injection injury effect, the brain slice from the injection side anterior to posterior 0.5 mm would not be used for quantification. The specific distance of the representative image is posterior 0.7 mm to the injection site. We also fixed the legend in Line 677.

Q12: Fig. 8/ Fig. S7 : As TDP-O Ab recognizes not only oligomers but also monomers (Ref. 18), the IR does not necessarily indicate only oligomers. Colocalization of TDP-43 with A β should be illustrated more clearly in figures using e.g. cytoplasmic markers of neurons.

A12: In the revised manuscript, we have performed triple staining with neuronal marker MAP2 as well as TDP-43 and A β . The result is described in text and data shown in Supplementary Fig. 8. The results showed that TDP-43 and A β majorly colocalized in neuronal cytoplasmic region.

Q13: Fig. S6 : Amyloid burden is not clear in the figures. Figures with higher contrast are required. Injection site should be indicated. As amyloid is detected in the olfactory tract (Line 304), figures at the level of the olfactory bulb are necessary.

A13: We thank the reviewer for this comment. We have updated the figure with a higher resolution and included the analysis for olfactory bulb (see Supplementary Fig. 7). The injection site is indicated in the method for “Intracranial injection to mice”. The results showed similar tendency that TDP-43 has higher amyloid burden in several brain regions.

Q14: As there are more than one APP/PS1 mouse line, notation of a more accurate

manner as APP/PS1deltaE9 or APP/PS1ΔE9 is better.

A14: We thank the reviewer to point out this confusion. We used the B6C3-Tg (APP^{swe},PSEN1^{dE9})85Dbo/Mmjax mice obtained from the Jackson Laboratory. To address this concern, we have stated the abbreviation for APP/PS1 mice to APP/PS1ΔE9 mice in the revised manuscript and SI.

Reviewer #2 (Remarks to the Author):

This is an interesting manuscript in which the authors have investigated the relationship between TDP-43 and amyloid-beta. The authors have found that TDP-43 inhibits amyloid-beta fibrillization at initial and oligomeric stages. Amyloid-beta fibrillization was delayed specifically in the presence of N-terminal domain containing TDP-43 40 variants, while C-terminal TDP-43 was not essential for amyloid-beta interaction. The authors also found that TDP-43 significantly enhances amyloid-beta ability to impair long-term potentiation and causes spatial memory deficit. These findings were consistent with results from experiments on Alzheimer's disease (AD) transgenic mice injected with TDP-43 primed amyloid-beta. Despite the topic of the manuscript is significant and might unravel important aspects of AD pathology, this reviewer has several concerns that reduce enthusiasm towards the overall manuscript. Specifically, the reviewer has the following criticisms:

Q1: a) Why have the authors performed aggregation experiments in vitro using Amyloid-beta 40 instead of oligomer prone amyloid-beta 42? One would have liked to see the results of the same experiments with the 42aa form of the peptide, as one cannot exclude that the longer peptide produces different results. Moreover, in the second part of the manuscript the authors used amyloid-beta 42 primed preparations of unknown characterization.

A1: We thank the reviewer for this comment. In the first submission, we have shown the inhibitory effect of TDP-43 variants on Aβ42 fibrillization in Supporting Figure 4. The ThT result also showed that full-length TDP-43 significantly inhibited Aβ42 fibrillization. Hence, Aβ42 used for the second part of the study has been characterized.

Q2: b) This reviewer is not clear on the fate of TDP-43 in the preparations in which

Amyloid-beta is primed with it. Has it been removed? Is it possible that the combination of amyloid-beta plus TDP-43 is responsible for the observed effects, instead of the modified amyloid-beta alone? Moreover, would it be possible that amyloid-beta and TDP-43 form mixed oligomers that are responsible for the observed effects?

A2: We are sorry for the confusion. TDP-43 was present in the *in vitro* preparations without being removed, but it is presented in only 1% (0.25 μ M) to A β (25 μ M) in molar ratio. Therefore, the effect should be majorly attributed from A β . However, indeed we cannot rule out the possible effect of TDP-43-A β complex from the 1% TDP-43. One will need to purify TDP-43-A β complex to address this specific question. But isolation of such complex can be difficult. We will work on it in the near future. To clarify the confusion, we have added explanation in discussion in the revised manuscript.

Q3: c) It is surprising to this reviewer that high concentrations of amyloid-beta (especially for the experiments performed with the 42aa peptide) alone do not impair LTP and behavior. One wonders the type of amyloid-beta used for the experiments shown on the manuscript. It does not seem that the recombinant amyloid-beta undergoes oligomerization through a method inspired by the classical method by Stine et al (J Biol Chem. 2003 Mar 28;278(13):11612-22)

A3: We thank the reviewer for pointing out the confusion. Previous publications have shown that A β 40 at 0.1-1 μ M incubation for 30 min did not affect LTP (Fig.3e and 5e, (Li, et al. 2018)), whereas, treatment of A β 42 at 1 μ M for 1 hr (Fig. 4, (Varga, et al. 2014)), 1 μ M for 24 hr (Fig. 3c, (Barucker, et al. 2015)), 0.5 μ M for 20 to 40 min (Fig.4, (Lauren, et al. 2009)), and 0.5 μ M for 2 hr (Fig. 3a, (Grochowska, et al. 2017) showed impairment of LTP.

In our LTP experiments, we focused on examining TDP-43 effect on A β . A β was prepared without further separation and the treatment was 30 min. We chose to use non-toxic A β concentration that does not affect LTP in our condition to show the detrimental effect from TDP-43. We used total A β 40 concentration at 1 μ M with or without TDP-43 at 0.01 μ M. For A β 42, we have tested different concentrations of A β 42 (see below figure) and confirmed that A β 42 at 0.25 μ M did not affect LTP in our condition. Hence, in A β 42 experiments we used A β 42 at 0.25 μ M and TDP-43 at 0.0025 μ M. Our A β concentration is lower than those reported in the literature that caused LTP impairment.

Q4: d) All the experiments on APP/PS1 mice lack controls in non-transgenic mice treated with vehicle and those treated with amyloid-beta and/or TDP-43. These controls should have been performed in interleaved experiments with those on transgenic mice.

A4: We thank the reviewer for the comment. We indeed have performed the wild type littermate control experiments. In the revised manuscript, we added the data for the littermate control in Fig 6 and supplementary figure 6.

Minor criticisms:

Q5: a) The manuscript does not specify the concentration used for amyloid-beta 40 and 42 in the behavioral experiments shown on figure 5. The text reports a concentration (25 uM) for amyloid-beta without defining the length of the peptide. Did the authors use the same concentration for amyloid-beta 40 and 42? Why, considering that amyloid-beta 42 is much more toxic than amyloid-beta 40? This reviewer is worried that with these high concentrations the authors are interfering with mechanisms other than those related to synaptic dysfunction (i.e. cell death).

A5: We are sorry for this confusion. We have revised the manuscript to indicate concentration for Aβ40 and Aβ42 are both 25 μM (~ 0.112 μg/μl, ~224 ng in 2 μl). Previous literature showed that the behavior deficit in mice is caused by injection of much higher Aβ concentrations or repeated injection of a similar Aβ concentration. For example, a single bilateral icv injection of 1 μg/μl (Sharma, et al. 2016), repeated bilateral injection of 0.2 μg/μl Aβ42 once daily for 4 consecutive days (Faucher, et al. 2016), repeated bilateral hippocampus injection of 0.2 μg/μl Aβ42 once daily for 6 consecutive days (Brouillette, Caillierez et al. 2012), and bilateral icv injection of 0.45 μg/μl (Nicole, et al. 2016). Therefore, our Aβ injection concentration is not too high comparing to the literature. We did not reduce Aβ42 concentration from Aβ40 since

our purpose is to examine TDP-43 effect on A β . Meanwhile, we also provided A β alone as our control.

In spatial memory behavior study *in vivo*, there could be cell death other than synaptic loss. Previously literature has shown that injection of 0.5 μ L 0.1 μ g/ μ L A β to 10-12 month WT mice induced 30% neuronal loss at 28 days after post-injection (Takuma, et al. 2004). Based on our result, we described that "TDP-43 induced A β significantly impaired spatial learning and memory of mice to a greater extent than A β or TDP-43 does" without specifying the cause since in this experiment we cannot differentiate synaptic dysfunction or neuronal loss. But, in the LTP experiment we did show that TDP-43-induced A β species significantly impairs synapse function to a greater extent than A β alone or TDP-43 did. We have revised the description to make it clearer.

Q6:b) One would have liked to have displayed findings from all quadrants in the probe trial of the Morris water maze test, to make it sure that the assay is properly performed.

A6: We thank the reviewer for this comment. We have drawn the data from all quadrants in the probe trial of Morris water maze test (see Fig 5c, Fig 5d, and Fig 6b).

Reviewer #3 (Remarks to the Author):

In the present paper by Shih et al. the authors studied the effect of TDP-43 on Amyloid-beta aggregation and have extended earlier work which showed that full-length TDP-43 forms oligomers and agitates fibrillization (Fang YS Nat Communications, 2014). The main finding of the present study is that TDP-43 inhibited and delayed A β fibrillization and oligomerization. Moreover, TDP-43 injection into an AD mouse model (APPPS1) increased A β plaque load and impaired spatial memory in those mice. The authors propose that TDP-43 inhibits A β fibrillization but exacerbates AD pathology. These findings are somehow contradictory. As detailed below, the authors over-interpret their findings and there are some technical concerns. I have specific comments that the authors can tackle in a straightforward revision.

Specific comments:

Q1: 1) The *in vitro* aggregation data of Figure 1 suggests, that there is a prolonged lag

phase in the presence of TDP-43. Furthermore the authors show that Ab fibrillization is reduced by TDP-43. On the other hand the authors show that amyloid plaque load was significantly increased in several brain regions upon TDG-43 injection into APPPS1 transgenic mice. These data are contradictory and need to be discussed in detail. It is not enough to mention in the discussion section on page 12 that 'The discrepancy among these studies indicates.....is complex.....and should all be considered'.

A1: We are sorry for the confusion. We revised the discussion in the current manuscript as below: Together, our studies showed that TDP-43 plays multiple roles in AD including induction of more A β oligomers through direct interaction with A β leading to synaptic dysfunction and memory impairment, increasing brain inflammation that also worsens spatial memory and lead to increase of amyloid burden (Flores, et al. 2018; Meraz-Rios, et al. 2013), and affecting APP metabolism (O'Brien and Wong 2011). These events could occur in the same time periods but affect AD pathology in different degrees.

Q2: 2) Although the second sentence in the discussion implies that 'TDP-43 oligomers affect a conformational change in A β thereby preventing the formation of Ab fibrils' – this has not been shown and it is an open question on how this should work (mechanism).

A2: We thank the reviewer for this comment. Actually, we have performed far-UV CD measurement to monitor A β conformational change along with the experiments described in Fig 1. In the revised manuscript, we added the time course CD spectra in Fig. 1c to show the conformational changes of A β 40 affected by the presence of TDP-43. The result is described in text. As for the mechanism, since we have shown that TDP-43 and A β interacts especially through RRM domains, we hypothesized that A β and TDP-43 interaction through RRM domains but require dimeric/oligomeric through N-terminal TDP-43 to inhibit A β fibrillization. The inhibition results in retention of A β oligomers. We have edited our discussion to make it clearer.

Q3: 3) It is not clear why TDP-43 impairs spatial memory? Is this due to more A β plaques? Please clarify and explain.

A3: Based on our results from the *in vitro* and *in vivo* studies, we proposed that TDP-

43 plays several roles leading to impairment of spatial memory including TDP-43 induced more A β oligomers (as evidenced by data in Fig. 2, Fig. 7, and supplementary Fig.10) as well as TDP-43 increased inflammation (Fig. 6f). Both A β oligomers and inflammation have been shown to impair special memory in mice(Flores, et al. 2018; Lesne, et al. 2006; Meraz-Rios, et al. 2013). In the revised manuscript, we have explained the role of TDP-43 in the discussion.

Q4: 4) Additional parameters should be used to control for the animals behavior in the pool such as test for visual ability and motivation (e.g. swim speed).

A4: We thank the reviewer for this comment. In our experiments, each mouse was subjected a two-day pre-training trials as described in methods that demonstrating their visual ability and spatial navigation motivation. In the revised manuscript, we have added the data for each mouse as well as their swimming velocity (see supplementary Fig. 5 and 6)

Q5: 5) The study would improve if the in vitro data would be repeated with rigorous in vivo seeding experiments in young pre-depositing APPPS1 transgenic mice that should be analyzed at different time points (time course). Due to the prolonged lag phase in vitro one would expect reduced Ab seeding. Those data should be more comparable to the in vitro data.

A5: We thank the reviewer for the suggestion. Since this study requires a large scale of animal study, we will try it in the future.

Q6: 6) Although the authors claim in the last sentence of the discussion that TDP-43 exacerbates neuronal damage of A β and AD related pathology they did not study or show neuronal damage at all. Therefore this sentence has to be at least rephrased.

A6: We thank the reviewer for pointing out this mistake. We have revised the sentence to “Overall, our results demonstrate that TDP-43 inhibits A β fibrillization through interaction with A β and exacerbates AD related pathology”.

References:

Barucker, C., et al.

2015 Abeta42-oligomer Interacting Peptide (AIP) neutralizes toxic amyloid-beta42 species and protects synaptic structure and function. *Sci Rep* 5:15410.

Faucher, Pierre, et al.

2016 Hippocampal Injections of Oligomeric Amyloid β -peptide (1–42) Induce Selective Working Memory Deficits and Long-lasting Alterations of ERK Signaling Pathway. *7*(245).

Flores, J., et al.

2018 Caspase-1 inhibition alleviates cognitive impairment and neuropathology in an Alzheimer's disease mouse model. *Nat Commun* 9(1):3916.

Grochowska, K. M., et al.

2017 Posttranslational modification impact on the mechanism by which amyloid-beta induces synaptic dysfunction. *EMBO Rep* 18(6):962-981.

Lauren, J., et al.

2009 Cellular prion protein mediates impairment of synaptic plasticity by

amyloid-beta oligomers. *Nature* 457(7233):1128-32.

Lesne, S., et al.

2006 A specific amyloid-beta protein assembly in the brain impairs memory. *Nature* 440(7082):352-7.

Li, S., et al.

2018 Decoding the synaptic dysfunction of bioactive human AD brain soluble Abeta to inspire novel therapeutic avenues for Alzheimer's disease. *Acta Neuropathol Commun* 6(1):121.

Meraz-Rios, M. A., et al.

2013 Inflammatory process in Alzheimer's Disease. *Front Integr Neurosci* 7:59.

Nicole, Olivier, et al.

2016 Soluble amyloid beta oligomers block the learning-induced increase in hippocampal sharp wave-ripple rate and impair spatial memory formation. *Scientific Reports* 6(1):22728.

O'Brien, R. J., and P. C. Wong

2011 Amyloid precursor protein processing and Alzheimer's disease. *Annu Rev Neurosci* 34:185-204.

Sharma, S., et al.

2016 Alzheimer's disease like pathology induced six weeks after aggregated amyloid-beta injection in rats: increased oxidative stress and impaired long-term memory with anxiety-like behavior. *Neurol Res* 38(9):838-50.

Takuma, H., et al.

2004 Amyloid beta peptide-induced cerebral neuronal loss is mediated by caspase-3 in vivo. *J Neuropathol Exp Neurol* 63(3):255-61.

Varga, E., et al.

2014 Abeta(1-42) enhances neuronal excitability in the CA1 via NR2B subunit-containing NMDA receptors. *Neural Plast* 2014:584314.

REVIEWER COMMENTS

Reviewer #1 (Remarks to the Author):

The manuscript has been much improved after the revision. Authors have appropriately responded to each of the reviewer's comments.

In the new suppl. Fig. 7 that includes additional data on OB, symbols for statistical significance at several regions changed from previous suppl. Fig. 6. Did the authors add samples for analysis?

As colocalization of A-beta with TDP-43 in the neuronal cytoplasm is important in the context of this manuscript, the reviewer suggests that figures beautifully illustrating the colocalization in suppl. Fig. 8 are to be moved to Fig. 8.

Reviewer #2 (Remarks to the Author):

The authors have partially addressed previous criticisms, but fundamental concerns are still present with this reviewer. Specifically, it is not clear why the inhibitory effect of TDP-43 variants on A β 42 fibrillization is different than those on A β 40 fibrillization. The explanation provided by the authors that the rapid aggregation of A β 42 might reduce the effective interaction between TDP-43 and A β 42 does not make sense given that one of the variants has a more striking effect on A β 42 than the normal TDP-43.

Additional criticisms are:

a) this reviewer is still concerned about the presence of TDP-43 in the preparation which might alter the interpretation of the results, nor the fact that TDP-43 is only 1% of the preparation alleviates concerns. Additionally, it is well known that the type of aggregation is profoundly influenced by the method of aggregation.

b) with respect to the criticism on the dose of A β 42 producing LTP reduction, 0.25 μ M normally impairs it, suggesting that the preparation of A β 42 is differently prepared than in other labs (please notice that the figure shown in the rebuttal does not show high quality results due to the upward drift of the baseline prior to the tetanus).

c) controls for the APP/PS1 mice should have been performed in interleaved experiments. Adding to the figure control data obtained separately is not appropriate (please also notice that data from the WT mice in the first trial are very different than those from APP/PS1 mice, suggesting that differences among mouse groups are not due to cognitive problems).

This reviewer suggests also improving the quality of the English language.

Reviewer #3 (Remarks to the Author):

The Authors have adequately addressed almost all the comments that have been raised in a previous round of review only the proposed in vivo seeding experiment was not examined.

We thank you for considering our manuscript entitled “**TDP-43 interacts with amyloid- β , inhibits fibrillization, and worsens Alzheimer’s disease pathology**”. After the first revision, here we response to the reviewers’ questions in the second revision letter point-by-point as listed below.

REVIEWER COMMENTS

Reviewer #1 (Remarks to the Author):

Comment: The manuscript has been much improved after the revision. Authors have appropriately responded to each of the reviewer’s comments.

In the new suppl. Fig. 7 that includes additional data on OB, symbols for statistical significance at several regions changed from previous suppl. Fig. 6. Did the authors add samples for analysis?

Response: In the first version, we quantified the amyloid plaque density in the olfactory bulb, anterior olfactory nucleus, and olfactory tubercle together as the olfactory tract. In the first revision, we did not add new samples for analysis. We re-quantified the olfactory bulb separately from the anterior olfactory nucleus and olfactory tubercle. The significance p-value changed due to the differences in number of comparing groups and the quantified values. The statistical methods are the same.

Comment: As colocalization of A-beta with TDP-43 in the neuronal cytoplasm is important in the context of this manuscript, the reviewer suggests that figures beautifully illustrating the colocalization in suppl. Fig. 8 are to be moved to Fig. 8.

Response: We thank the reviewer for the comment. In this revision, we have moved Suppl. Fig.8 to Figure 9 in text.

Reviewer #2 (Remarks to the Author):

Comment: The authors have partially addressed previous criticisms, but fundamental concerns are still present with this reviewer. Specifically, it is not clear why the inhibitory effect of TDP-43 variants on A β 42 fibrillization is different than those on A β 40 fibrillization. The explanation provided by the authors that the rapid aggregation of A β 42 might reduce the effective interaction between TDP-43 and A β 42 does not make sense given that one of the variants has a more striking effect on A β 42 than the normal TDP-43.

Response: We understood the reviewer’s concern on the differences in A β 40 and A β 42, but we were confused with the comment on “one of the variants has a more striking effect on A β 42 than the normal TDP-43”. In our data in S4, only full-length (normal) TDP-43 inhibit A β 42. Therefore, there is no other variant that has a more striking effect on A β 42. With this concern, we determined the T₅₀ (the time required to reach half of the fluorescence enhancement) for each kinetic data and provided statistical analysis below to show that only full-length TDP-43 had more profound inhibitory effect but not the variants on A β 42 aggregation.

For the concern on the difference in A β 40 and A β 42, we thank the reviewer for bringing up the question. In our biochemical studies we have kept the sample preparation and experimental condition tightly controlled. We found that A β 42 aggregation is much faster than A β 40 which is consistent with many other studies¹⁻³. In the A β 40 experiments, we found TDP-43 can only inhibit A β 40 monomers and oligomers (Fig. 1). Therefore, it is likely that A β 42 cannot be effectively inhibited by TDP-43 due to A β 42 forming several oligomers at the beginning with less monomer^{1,4,5} and undergo fast fibrillization. This is why we speculate that TDP-43 cannot effectively inhibit A β 42 fibrillization due to rapid aggregation of A β 42. Furthermore, A β 40 and A β 42 oligomerize through distinct pathways^{1,4,5}. Freshly prepared A β 40 predominantly forms monomer¹, forms a monomer-added mixture after crosslinking⁴, and undergoes slow fibrillization process¹⁻³, whereas, freshly prepared A β 42 forms paranuclei^{4,5} or tetramer^{1,5} in equilibrium with monomer, then undergoes fast aggregation. Therefore, the potential interaction site and species of A β 40 and A β 42 for TDP-43 can be different. It requires further intensive biophysical investigation to reveal the exact mechanism. We added more detailed discussion on these possibilities in the second revised manuscript.

Comment: Additional criticisms are:

a) this reviewer is still concerned about the presence of TDP-43 in the preparation which might alter the interpretation of the results, nor the fact that TDP-43 is only 1% of the preparation alleviates concerns. Additionally, it is well known that the type of aggregation is profoundly influenced by the method of aggregation.

Response: We thank the reviewer for bringing out this concern again. We do carefully think about the possible ways to separate 1% TDP-43 from A β aggregation. However, it is technically difficult to separate species in an aggregation reaction since TDP-43 and A β both form oligomers. Even if we perform possible immunoprecipitation or size-exclusion chromatography, the process that changes pH or protein concentration may change the species in the reaction. Also, A β isolated from TDP-43 may adopt A β alone conformation and lose the TDP-43 effect. For example, in our previous study ⁶, we found that Zn ions induce A β to form stable and toxic oligomers, however, once Zn ions were removed by EDTA, A β immediately changed back to A β alone conformation and start to form amyloid fibrils. Based on these reasons, we hope the reviewer understand there is a technical difficulty to separate a defined aggregated species and maintain the original property in the currently research field.

Comment: b) with respect to the criticism on the dose of A β 42 producing LTP reduction, 0.25 μ M normally impair it, suggesting that the preparation of A β 42 is differently prepared than in other labs (please notice that the figure shown in the rebuttal does not show high quality results due to the upward drift of the baseline prior to the tetanus).

Response: We thank the reviewer for the comment. We believe the discrepancy comes from the differences in the sample source/preparation procedure and protein quantification method. Here, we compared our condition with the other studies.

In a study for LTP impairment by A β 42 oligomers ⁷, A β 42 was prepared from the lyophilized peptide, resuspend to 1 mM in HFIP. After removal of HFIP by evaporation, A β was dissolved in DMSO for 10 mins in sonication and vortexed for 30 sec, then incubated for 24 hr at 4 °C. It was further diluted by artificial CSF to get 200 nM A β 42 oligomers. A β 42 oligomers at 200 nM were able to impair LTP. A similar protocol for A β 42 oligomer preparation but in F12 medium showed LTP impairment at 500 nM after treatment for 20-40 min ⁸ and 2 hr ⁹. In another study,

A β 42 was lyophilized in trifluoroacetic acid, replaced by 2 mM HCl, dissolved in PBS, and diluted to 100 μ M in HEPES buffer after 2 day incubation at room temperature. Under this condition, A β 42 impaired LTP at 1 μ M, but not at 0.1-0.5 μ M¹⁰.

In addition, the quantification method as well as the source for A β were different in each study. The method chosen to determine A β concentration was dependent on the experimental design and limitation¹¹. Most studies calculated A β 42 concentration based on the molecular weight of the peptide without re-quantification or further indication^{8-10,12,13}. Some studies used the commercial ELISA kit to obtain A β 42 concentration. In these studies, the LTP would be impaired under 0.7-1 μ M of A β 42 treatment^{14,15}. In a study using absorbance at 275 nm to determine synthetic A β concentration, A β 42 was dissolved in 7 M GndHCl and 5 mM ethylenediaminetetraacetic acid and incubated overnight. The LTP would be impaired under 0.2-0.5 μ M of the A β 42 treatment¹⁶. In these studies, A β 42 were either synthesized from a laboratory¹² or purchased from various companies including American peptide^{7,8,13}, AnaSpec^{9,10}, and California Peptide⁸. Peptide synthesis of A β from different sources have also been known to affect its properties. Therefore, different A β preparation could lead to different results in LTP impairment.

Here, we dissolved our lyophilized A β 42 peptide in 10 mM Tris buffer and re-quantified A β concentration by micro BCA assay as suggested by Nature Protocol 2010. We further incubated the samples for ~40 hr for aggregation and treated the brain slice for 30 min with A β 42 at 0.25 μ M. We cannot compare our result with the others due to differences in sample preparation/source and concentration determination. As for the quality of the figure in the first response letter, we did not intend to publish the figure. It is a pilot test for A β 42 concentration determination on LTP study.

Comment: c) controls for the APP/PS1 mice should have been performed in interleaved experiments. Adding to the figure control data obtained separately is not appropriate (please also notice that data from the WT mice in the first trial are very different than those from APP/PS1 mice, suggesting that differences among mouse groups are not due to cognitive problems).

Response: We totally agree with the reviewer and indeed we have performed wild-type littermate experiments side-by-side in parallel with APP/PS1. It is our stupidity not to show wild-type littermate result in our first submission. In the first revised manuscript, we have added the wild-type littermate data. They are indeed interleaved

experiments. Here, we attached the video files with the established time to prove all of the wild-type littermate mice and APP/PS1 mice were performed at the same time. For example, there are 32 video files established on the same day in the probe test folder (please reference the original video file we provide, the time list has been attached as the excel file).

Probe test video can be checked with follow

link: https://drive.google.com/drive/folders/1eWctD3qtLuifN9YYRvj_nvMFHg6M2Fl?usp=sharing

Other behavior video can be checked with follow

link: <https://drive.google.com/drive/folders/1tWszK8LUzOqX9M4D-NR-jDg12R3SmXkT?usp=sharing>

Comment: This reviewer suggests also improving the quality of the English language.

Response: We have sent our first submission file and the currently revised manuscript to professional English editing. We have made changes accordingly in this revision.

Reviewer #3 (Remarks to the Author):

The Authors have adequately addressed almost all the comments that have been raised in a previous round of review only the proposed in vivo seeding experiment was not examined.

Response: We thank the reviewer for the comment.

With the above point-by-point answers to the reviewers' questions, I believe we now addressed all inquiries and our manuscript is ready to be published in *Nature Communications*. Thank you very much!

Sincerely yours,

Yun-Ru Chen, Ph.D.
Associate Professor
Genomics Research Center
Academia Sinica, Taipei, Taiwan
Tel: 886-2-2787-1275,
E-mail: yrchen@gate.sinica.edu.tw

References

- 1 Chen, Y. R. & Glabe, C. G. Distinct early folding and aggregation properties of Alzheimer amyloid-beta peptides A beta 40 and A beta 42 - Stable trimer or tetramer formation by A beta 42. *Journal of Biological Chemistry* **281**, 24414-24422 (2006).
- 2 Fezoui, Y. & Teplow, D. B. Kinetic studies of amyloid beta-protein fibril assembly. Differential effects of alpha-helix stabilization. *J Biol Chem* **277**, 36948-36954 (2002).
- 3 Kuperstein, I. *et al.* Neurotoxicity of Alzheimer's disease Abeta peptides is induced by small changes in the Abeta42 to Abeta40 ratio. *EMBO J* **29**, 3408-3420 (2010).
- 4 Bitan, G. *et al.* Amyloid beta -protein (Abeta) assembly: Abeta 40 and Abeta 42 oligomerize through distinct pathways. *Proc Natl Acad Sci U S A* **100**, 330-335 (2003).

- 5 Bernstein, S. L. *et al.* Amyloid- β protein oligomerization and the importance of tetramers and dodecamers in the aetiology of Alzheimer's disease. *Nature chemistry* **1**, 326-331 (2009).
- 6 Lee, M. C. *et al.* Zinc ion rapidly induces toxic, off-pathway amyloid-beta oligomers distinct from amyloid-beta derived diffusible ligands in Alzheimer's disease. *Scientific reports* **8**, 4772 (2018).
- 7 Puzzo, D. *et al.* Amyloid-beta peptide inhibits activation of the nitric oxide/cGMP/cAMP-responsive element-binding protein pathway during hippocampal synaptic plasticity. *J Neurosci* **25**, 6887-6897 (2005).
- 8 Lauren, J., Gimbel, D. A., Nygaard, H. B., Gilbert, J. W. & Strittmatter, S. M. Cellular prion protein mediates impairment of synaptic plasticity by amyloid-beta oligomers. *Nature* **457**, 1128-1132 (2009).
- 9 Grochowska, K. M. *et al.* Posttranslational modification impact on the mechanism by which amyloid-beta induces synaptic dysfunction. *EMBO reports* **18**, 962-981 (2017).
- 10 Nakagami, Y. *et al.* A novel beta-sheet breaker, RS-0406, reverses amyloid beta-induced cytotoxicity and impairment of long-term potentiation in vitro. *British journal of pharmacology* **137**, 676-682 (2002).
- 11 Jan, A., Hartley, D. M. & Lashuel, H. A. Preparation and characterization of toxic A β aggregates for structural and functional studies in Alzheimer's disease research. *Nature protocols* **5**, 1186-1209 (2010).
- 12 Nicoll, A. J. *et al.* Amyloid-beta nanotubes are associated with prion protein-dependent synaptotoxicity. *Nat Commun* **4**, 2416 (2013).
- 13 Hughes, C. *et al.* Beta amyloid aggregates induce sensitised TLR4 signalling causing long-term potentiation deficit and rat neuronal cell death. *Commun Biol* **3**, 79 (2020).
- 14 Varga, E. *et al.* A β (1-42) enhances neuronal excitability in the CA1 via NR2B subunit-containing NMDA receptors. *Neural Plast* **2014**, 584314 (2014).
- 15 Fulop, L. *et al.* A foldamer-dendrimer conjugate neutralizes synaptotoxic beta-amyloid oligomers. *PLoS One* **7**, e39485 (2012).
- 16 Li, S. *et al.* Decoding the synaptic dysfunction of bioactive human AD brain soluble A β to inspire novel therapeutic avenues for Alzheimer's disease. *Acta neuropathologica communications* **6**, 121 (2018).

REVIEWER COMMENTS

Reviewer #2 (Remarks to the Author):

The responses to my previous critiques are perplexing for the following reasons:

- a) data for Amyloid-beta 40 are different from the 42 amino acid peptide. Thus, conclusion cannot be extrapolated to amyloid-beta in general. This is not clear in the manuscript. This is important as the 42 amino acid peptide is the one that has been more often associated with disease pathology. There is the need to clarify better the differences between the two, to avoid adding confusion to available literature.
- b) the criticism on the dose of A β 42 producing LTP reduction is still valid, as a weak data cannot be used as an explanation for something, nor the fact that the data is not going to be shown reassures this reviewer that the criticism has been addressed.
- c) it is not possible for this reviewer to understand based on the file name shown in the rebuttal what files stand for.

In addition, this reviewer noticed that only male animals were used to perform the experiments. Use of females in addition to males is recommended.

In our third revision, we have performed new experiments and added new data to the revised manuscript. The responses to reviewer 2's questions are listed point-by-point below.

REVIEWER COMMENTS

Reviewer #2 (Remarks to the Author):

The responses to my previous critiques are perplexing for the following reasons:

- a) data for Amyloid-beta 40 are different from the 42 amino acid peptide. Thus, conclusion cannot be extrapolated to amyloid-beta in general. This is not clear in the manuscript. This is important as the 42 amino acid peptide is the one that has been more often associated with disease pathology. There is the need to clarify better the differences between the two, to avoid adding confusion to available literature.

Response: We thank the reviewer for this concern. Although our *in vitro* study is mainly based on A β 40 due to the sample handling problem, we have provided evidences to show TDP-43-induced A β 40 and A β 42 toxicity in LTP *ex vivo* (**Fig. 4**) and in water maze in the injection WT mice model *in vivo* (**Fig. 5**). We also showed both A β 40 and A β 42 colocalized in brain of post mortem AD patients. (**Supplementary Fig 9**). In the revised manuscript, we have added more description in both introduction and discussion to raise the attention to the possible differences resided in A β 40 and A β 42.

- b) the criticism on the dose of A β 42 producing LTP reduction is still valid, as a weak data cannot be used as an explanation for something, nor the fact that the data is not going to be shown reassures this reviewer that the criticism has been addressed.

Response: To provide valid data, we re-performed the LTP experiments for A β 42 and also prepared A β 42 according to previous literature to be consistent with the previous studies¹⁻³. The dose-dependent toxicity of A β 42 was measured and analyzed (See **Supplementary Fig. 5**). The result is described in the revised manuscript showing that A β 42 impaired LTP in a dose-dependent manner. To understand TDP-43 effect on A β 42, we diluted the samples to a non-toxic A β concentration at 62.5 nM to minimize A β 42 toxicity. The same molar ratio of TDP-43 (0.625 nM) to A β 42 (62.5 nM) at 1% was kept. Similarly, TDP-43-induced A β 42 significantly inhibited hippocampal LTP compared with that of the buffer control (see revised **Fig. 4b**). No significant

difference was found in the fEPSP slope among the buffer control, 62.5 nM A β 42, and 0.625 nM TDP-43. The result indicated that TDP-43-induced A β 42 species significantly impair synapse function to a greater extent than A β 42 alone or TDP-43 alone did.

- b) it is not possible for this reviewer to understand based on the file name shown in the rebuttal what files stand for. In addition, this reviewer noticed that only male animals were used to perform the experiments. Use of females in addition to males is recommended.

Response: Sorry to confuse the reviewer. Our intention is to prove our experiments were performed at the same time since the computer files showed the time information. We will consider female mice experiment in the future.

References

- 1 Puzzo, D. *et al.* Amyloid-beta peptide inhibits activation of the nitric oxide/cGMP/cAMP-responsive element-binding protein pathway during hippocampal synaptic plasticity. *J Neurosci* **25**, 6887-6897 (2005).
- 2 Lauren, J., Gimbel, D. A., Nygaard, H. B., Gilbert, J. W. & Strittmatter, S. M. Cellular prion protein mediates impairment of synaptic plasticity by amyloid-beta oligomers. *Nature* **457**, 1128-1132 (2009).
- 3 Grochowska, K. M. *et al.* Posttranslational modification impact on the mechanism by which amyloid-beta induces synaptic dysfunction. *EMBO reports* **18**, 962-981 (2017).

REVIEWERS' COMMENTS

Reviewer #2 (Remarks to the Author):

The authors have addressed my criticisms.

We thank all reviewers for their critical comments and suggestions. The response to reviewer 2 is below.

REVIEWERS' COMMENTS

Reviewer #2 (Remarks to the Author):

The authors have addressed my criticisms.

Response: We thank the reviewer for the comment.